# NitroNet - A deep-learning $NO_2$ profile retrieval prototype for the TROPOMI satellite instrument

Leon Kuhn[1, 2], Steffen Beirle[2], Sergey Osipov[2, 3], Andrea Pozzer[2], and Thomas Wagner[1, 2]

[1]Institute for Environmental Physics, University of Heidelberg, Germany
[2]Satellite Remote Sensing Group, Max-Planck Institute for Chemistry, Mainz, Germany
[3]King Abdullah University of Science and Technology, Thuwal, Saudi Arabia

**Correspondence:** Leon Kuhn (l.kuhn@mpic.de)

**Abstract.**

We introduce "NitroNet", a deep learning model for the prediction of tropospheric $NO_2$ profiles from satellite column measurements. NitroNet is a neural network, which was trained on synthetic $NO_2$ profiles from the regional chemistry and transport model WRF-Chem, operated on a European domain for the month of May 2019. This WRF-Chem simulation was

constrained by in-situ and satellite measurements, which were used to optimize important simulation parameters (e.g. the boundary layer scheme). The NitroNet model receives vertical $NO_2$ column densities (VCDs) from the TROPOMI satellite instrument and ancillary variables (meteorology, emissions, etc.) as input, from which it reproduces $NO_2$ concentration profiles. Training of the neural network is conducted on a filtered dataset, meaning that $NO_2$ profiles with strong disagreement (> 20 %) to colocated TROPOMI column measurements are discarded.

We present a first evaluation of NitroNet on a variety of geographical and temporal domains (Europe, US west coast, India, and China) and different seasons. For this purpose, we validate the $NO_2$ profiles predicted by NitroNet against satellite, in-situ, and MAX-DOAS measurements. The training data were previously validated against the same datasets. During summertime, NitroNet shows small biases and strong correlations to all three datasets (bias $= +6.7$ % and $R = 0.95$ for TROPOMI $NO_2$ VCDs, bias $= -10.5$ % and $R = 0.75$ for AirBase surface concentrations, bias $= -34.3$ % to $+99.6$ % and $R = 0.83$ to

0.99 for MAX-DOAS measurements). In the comparison to TROPOMI satellite data, NitroNet even shows significantly lower errors and stronger correlation than a direct comparison with WRF-Chem numerical results. During wintertime considerable low biases arise, because the summertime/late spring training data is not fully representative of all atmospheric wintertime characteristics (e.g. longer $NO_2$ lifetimes). Nonetheless, the wintertime performance of NitroNet is surprisingly good, and comparable to that of classic regional chemistry and transport models. NitroNet can demonstrably be used outside the ge-

ographic and temporal domain of the training data with only slight performance reductions. What makes NitroNet unique compared to similar existing deep learning models is the inclusion of synthetic model data, which has important benefits: Due to the lack of $NO_2$ profile measurements, models trained on empirical datasets are limited to the prediction of surface concentrations learned from in-situ measurements. NitroNet, however, can predict full tropospheric $NO_2$ profiles. Furthermore, in-situ measurements of $NO_2$ are known to suffer from biases, often larger than +20 %, due to cross sensitivities to photooxidants,

which other models trained on empirical data inevitably reproduce.

# 1   Introduction

Nitrogen oxides ($NO_x$ = NO + $NO_2$) are an important marker of air pollution. The negative impact of $NO_2$ on human health has been widely recognized (see e.g. Faustini et al. (2014); Mills et al. (2015); Chowdhury et al. (2021)). In many European countries, the recommended annual-average exposure limit of 10 $\mu$g m$^{-3}$ (see World Health Organization (2021)) is exceeded continuously. Active monitoring of tropospheric $NO_2$ is a crucial step in identifying pollution hotspots, localizing emissions, and designing long-term solutions to the pollution problem. Different $NO_2$ measuring methods exist. Many countries across the world deploy in-situ measurements at the surface (see e.g. the AirBase network, European Environment Agency). The Tropospheric Monitoring Instrument (TROPOMI, see Veefkind et al. (2012)) yields measurements of the tropospheric $NO_2$ vertical column density (VCD) with daily near global coverage and a ground pixel size of up to 3.5 km $\times$ 5.5 km. Lastly, ground-based MAX-DOAS measurements ("multi-axis differential optical absorption spectroscopy", see Platt and Stutz (2008); Hönninger et al. (2004)) are used to obtain tropospheric $NO_2$ profiles in a few selected places, by means of scanning the troposphere at different elevation angles. Although further measuring platforms (e.g. sondes, aircraft) and methods (e.g. Light Detection and Ranging instruments (LIDAR), or "cloud-slicing") exist, these are not routinely deployed (see e.g. Sluis et al. (2010); Bourgeois et al. (2022); Lange et al. (2023); Riess et al. (2023); Volten et al. (2009); Berkhout et al. (2018); Su et al. (2021), Marais et al. (2021)). Particularly aircraft measurements and cloud slicing are appreciated for resolving along the vertical axis, although at lower spatio-temporal resolutions (e.g. cloud slicing: seasonal means with 1° $\times$ 1° horizontal resolution and 5 tropospheric layers, see Marais et al. (2021)) or sparse spatio-temporal coverage (aircraft measurements). Altogether, these measurements are valuable for the quantification of tropospheric vertical column densities, surface concentrations, and to some extent the tropospheric profile shapes. Nonetheless, the described methods also have drawbacks:

– The TROPOMI instrument can measure the tropospheric column density, but it cannot resolve along the lightpath or the vertical axis, meaning it can principally not return vertical $NO_2$ profiles. Furthermore, the TROPOMI $NO_2$ VCD retrieval depends on a priori profiles. In the operational TROPOMI processor, these are taken from the TM5-MP model (see Krol et al. (2005)), whose low horizontal resolution of 1° $\times$ 1° is known to be one of the main causes of significant negative biases of typically $-10$ to $-20$ % (see Ialongo et al. (2020); Tack et al. (2021); Liu et al. (2021); Douros et al. (2023)), but in some cases even up to $-50$% (Lange et al. (2023)). Alternative data products with higher resolved a priori profiles exist, but are not available globally.

– In-situ measurements often utilize the molybdenum-based chemiluminescence method, which is known for its severe cross sensitivities to other atmospheric oxidants, causing large biases in the reported $NO_2$ concentrations (see Dunlea et al. (2007); Steinbacher et al. (2007); Lamsal et al. (2008); Boersma et al. (2009); Villena et al. (2012)). These biases typically range from $+20$ % to $+100$ %, but Villena et al. (2012) even report biases of up to $+300$ % in extreme cases. As described in detail further into the manuscript, these biases can be strongly reduced down to a few percent within our model framework.

- MAX-DOAS measurements are quite sparsely located and cannot provide dense spatial coverage. Additionally, the commonly used retrieval algorithms suffer from significantly reduced sensitivity at higher altitudes ($> 2\,\mathrm{km}$), and depend on a priori assumptions. An intercomparison study of MAX-DOAS retrieval algorithms by Tirpitz et al. (2021) revealed relative retrieval uncertainties of between 3 % and 70 %, which can be expected to be the dominant part of the total MAX-DOAS uncertainty.

Measurements are therefore often complemented by regional chemistry and transport (RCT) simulations. Examples of state-of-the-art RCT models are WRF-Chem (Grell et al. (2005)), COSMO/MESSy (Kerkweg and Jöckel (2012)), Lotos-Euros (Manders et al. (2017)), CAM-chem (Emmons et al. (2020)), and CHIMERE (Menut et al. (2021)). Such models can simulate realistic distributions of $NO_2$ and other atmospheric trace gases with horizontal resolutions on the scale of $3\,\mathrm{km} \times 3\,\mathrm{km}$ and vertical resolutions of $\sim 1\,\mathrm{m}$ at the surface, to $\sim 1\,\mathrm{km}$ in the upper troposphere. High-resolution RCT simulations can be used to estimate air pollution in the absence of in-situ measurement, and to obtain better resolved a priori profiles for the TROPOMI retrieval. Unfortunately, the continuous deployment of RCT simulations is no easy endeavour, due to their computational expense, dependence on input data which may not always be available in an up-to-date form at high resolution (in particular emission data), and the uncertainty in choice of simulation parametrizations. Another point of concern is the general accuracy of these models: RCT simulations reported in recent literature have shown significant deviations from observational reference data (see Visser et al. (2019); Kuik et al. (2016); Kuik et al. (2018); Poraicu et al. (2023)), e.g. an underestimation of the summertime surface-level $NO_2$ concentration of up to $-50$ %. A study by Douros et al. (2023) reveals overestimations of the winter-time $NO_2$ VCD by $+50$ %, and demonstrates that such biases even occur in ensemble models, such as the Copernicus Atmosphere Monitoring Service regional model (CAMS regional, consisting of 11 different RCT models with $0.1° \times 0.1°$ horizontal resolution). In previous work, we showed that a recalibration of the vertical mixing parametrization can mostly resolve such biases in the WRF-Chem model in summer over Europe (see Kuhn et al. (2024)). However, the process of model recalibration is tedious, computationally expensive, and domain-dependent. Altogether, it can be concluded that high-resolution RCT simulations are of undisputed benefit, but their practical realization remains challenging.

In this article we introduce "NitroNet", a new machine learning model intended to complement existing RCT models and measurements of $NO_2$. NitroNet is a feed-forward neural network, which was designed to predict full tropospheric $NO_2$ profiles using TROPOMI VCDs alongside other ancillary data (meteorology, emissions, surface types, etc.) as input. Because neural networks are universal function approximators, they are the ideal tool to capture such complex data relationships. NitroNet is trained on numerically simulated data from the WRF-Chem model, operated on a European domain for the month of May 2019 as described in Kuhn et al. (2024). A data filtering scheme is used to ensure that only well-validated results from the WRF-Chem simulation are used for training the neural network, e.g. training examples with significant disagreement to colocated satellite observations are dismissed. Afterwards, NitroNet is used as a standalone model, without the necessity to run the RCT simulation again. NitroNet expands on previous deep learning models trained on empirical data (see e.g. Gardner and Dorling (1999); Kang et al. (2021); Chan et al. (2021); Ghahremanloo et al. (2021); Zhang et al. (2022); Jesemann et al. (2022); Cao (2023); all presented models were trained on in-situ surface observations) by inclusion of synthetic model data. This approach provides intrinsic advantages: Firstly, NitroNet can predict full $NO_2$ profiles, while models trained on empirical

data can only be used for surface predictions. Secondly, the chemical mechanisms of RCT models allow for the explicit treatment of in-situ measurement biases (typically larger than $+20$ %) by computation of suitable correction factors, while empirically trained models cannot compute such correction factors and inevitably reproduce the biases inherent to the training data. Thirdly, synthetic datasets of $NO_2$ profiles are typically much larger than the few empirical data, and also cover the spatial domain continuously. This allows for the use of highly selective training data filtering, which demonstrably improves the neural network's performance.

The article is structured as follows: Section 2 gives an overview of the datasets used in our study. Section 3 gives a detailed explanation of the NitroNet model. Section 4 shows an evaluation of NitroNet against satellite, in-situ, and MAX-DOAS data on a European domain for May 2022 (i.e. on input data, which the neural network has never seen before). This study is then extended to different seasons and geographical domains (UK, Spain + Portugal, US west coast, India, and China). Section 5 concludes.

## 2 Datasets

The following datasets are used in our study:

### 2.1 Vertical $NO_2$ profiles from WRF-Chem

An RCT simulation using the WRF-Chem model (v. 4.2.2, see Grell et al. (2005)) provides the $NO_2$ profiles on which NitroNet is trained. The simulation was run for the month of May 2019 on a domain over Europe with a spatial resolution of $3\,\mathrm{km} \times 3\,\mathrm{km}$, 43 terrain-following pressure levels, and hourly output. A detailed description, discussion, and validation study of this dataset was published in Kuhn et al. (2024). This study revolved around the question, how certain WRF-Chem model parameters can be optimized in order to improve the model's agreement to various reference datasets. In particular, optimization of the model's vertical mixing parametrization was found to be crucial to improve the agreement to in situ observations of surface $NO_2$. Unfortunately, such optimization processes take a long time to solve if the underlying model is as computationally expensive as WRF-Chem. Additionally, wintertime RCT simulations are known to be particularly challenging (see e.g. Douros et al. (2023)), mainly due to their tendency to overestimate the total $NO_2$ columns severely. Therefore, full-year training data with a resolution and accuracy compared to our summertime data cannot be provided for now. Although NitroNet was trained exclusively on summertime data, it can be used in other seasons as well, although with larger prediction errors (as discussed in sect. 4.3).

The simulation setup additionally deploys the vertical emission profiles from Bieser et al. (2011). We will refer to this dataset as "WRF-2019" from hereon. WRF-2019 contains approximately two million $NO_2$ profiles, which are split into three partitions: A training set (80 %), a validation set (15 %), and a test set (5 %). The training set is used for training NitroNet (described in sect. 3.3), the validation set for hyperparameter optimization (described in sect. 3.2 and Appendix A), and the test set for evaluation of the neural network on previously unseen data. The partitioning is obtained by unweighted random sampling without replacement.

## 2.2 Input data for NitroNet

NitroNet uses tropospheric $NO_2$ vertical column densities (VCDs) from the TROPOMI satellite instrument as the main input. Additionally, although much less influential, total $O_3$ VCDs are used, assuming they are informative of the tropospheric $O_3$ column, and thus of the tropospheric $NO_x$ photochemistry. The TROPOMI instrument on board of the S5P satellite observes spectra of backscattered light from space with near global coverage, a daily overpass at around 13:30 local time, and a pixel size of up to $3.5 \times 5.5$ km (see Veefkind et al. (2012); van Geffen et al. (2022)). The retrieval of tropospheric $NO_2$ VCDs is comprised of three steps: First, the $NO_2$ total slant column density is obtained from the observed light spectra using differential optical absorption spectroscopy (DOAS, see Platt and Stutz (2008)). Then, the obtained total SCD is separated into a stratospheric and a tropospheric component ($SCD_{trop}$). Finally, the tropospheric VCD is obtained by computing

$$VCD_{trop} = \frac{SCD_{trop}}{AMF_{trop}} \tag{1}$$

where $AMF_{trop}$ denotes the tropospheric air mass factor. Air mass factors are computed using an altitude-dependent look-up table together with simulated $NO_2$ a priori profiles from the RCT model TM5-MP (see Krol et al. (2005)) with a horizontal resolution of $1° \times 1°$. The process is described by van Geffen et al. (2022). Throughout our study, we only use data with a high "quality assurance" value ($f_{QA} > 0.75$), which is the general recommendation (see Eskes et al. (2019)). This also acts as a cloud filter, as it removes observations with cloud fractions of above 50 %. Throughout the rest of the paper, "$NO_2$ VCD" refers to the *tropospheric* $NO_2$ VCD, and "$O_3$ VCD" refers to the *total* $O_3$ VCD.

Additionally, NitroNet uses meteorological variables from the ERA5 reanalysis ($0.25° \times 0.25°$, see Hersbach et al. (2020)) and emission data from the EDGARv5 global emission inventory ($0.1° \times 0.1°$, see Crippa et al. (2020)) as input data.

## 2.3 Validation data for NitroNet

The following three datasets are used to evaluate the NitroNet model:

1. The aforementioned tropospheric $NO_2$ VCDs from the TROPOMI satellite instrument.

2. In-situ surface measurements of $NO_2$ from the European AirBase instrument network (see European Environment Agency). This dataset is assembled from the submissions of individual countries of the European Union. The measurements are available as hourly mean values and are classified into three groups: background, traffic, and industrial. Traffic and industrial stations are typically located directly next to strong sources (e.g. near large streets or power plants), where strong horizontal $NO_2$ gradients occur on the scale of a few meters (see e.g. Beckwith et al. (2019)). Such gradients can neither be resolved by TROPOMI, whose observations are used as input data, nor by WRF-Chem, whose simulation results were used for training NitroNet. Therefore, only background stations are included in our validation study.

3. $NO_2$ concentration profiles from MAX-DOAS instruments, operated within the "Fiducial Reference Measurements for Ground-Based DOAS Air-Quality Observations" project in Europe (FRM$_4$DOAS, see Fayt et al. (2021)). FRM$_4$DOAS

uses the optimal-estimation based Mexican MAX-DOAS fit (MMF, see Friedrich et al. (2019)), and the Mainz Profile Algorithm (MAPA, see Beirle et al. (2019)) for profile inversion. The resulting $NO_2$ profiles are defined on a vertical grid with approximately $\sim 200$ m spacing, reaching to altitudes of up to 4 km. Each instrument produces approximately five $NO_2$ profiles per hour. All profiles flagged as "erroneous" by MAPA were discarded. Note, that although MAPA does not support automatic cloud filtering yet, the described "error" flagging was shown to be sensitive to cloud effects, as well (see Beirle et al. (2019)).

## 3 NitroNet model description

The NitroNet model consists of an artificial neural network at its core and deploys additional non-machine learning code for efficient data pre-processing and Monte Carlo uncertainty estimation on high performance computing (HPC) architectures. NitroNet's neural network uses the feed-forward topology and is trained with the standard backpropagation method (see Rumelhart et al. (1986)). It has one output neuron, which is used to predict a single $NO_2$ concentration value per query. Full $NO_2$ profiles are obtained by concatenating multiple queries on a vertical grid of the user's choice. Although WRF-2019 is resolved on 43 vertical pressure levels, these correspond to different altitudes above ground across the spatio-temporal model domain. Therefore, NitroNet can be trained to predict the $NO_2$ concentration at arbitrary tropospheric altitudes. Throughout this article, a vertical grid with 186 levels is used, resulting in vertical resolutions of $\sim 1$ m near the surface, $\sim 50$ m up until 4 km altitude, and up to $400$ m in the regions between 4 and 8 km altitude.

### 3.1 Description of the model input

The purpose of our model is to provide realistic $NO_2$ profiles without the necessity to run computationally expensive RCT simulations. For this reason it is imperative, that NitroNet is only trained on variables from sources accessible both at training and runtime. This may include simulation data from other operational models (e.g. the planetary boundary layer height (PBLH) from ERA5), but excludes many potentially informative variables exclusive to WRF-2019 (e.g. various trace gas concentrations). The training targets (i.e. the $NO_2$ profiles) are exempt from this rule, because they can only be obtained from WRF-Chem. In contrast to sect. 2.2, the descriptions given here are based on our design choices, e.g. how the used data were selected and processed.

Table 1 gives an overview of all input variables ("features") to the neural network. For the $NO_2$ and $O_3$ VCDs, the most recent TROPOMI product version (2.04) is used. Tropospheric averaging kernels (AKs) are computed according to Eskes et al. (2019) and defined on the vertical grid of the TM5 model. NitroNet uses the tropospheric AKs at the 9 lowest TM5 layers (reaching up to $\sim 2300$ m altitude), although in hindsight, it was discovered that the AKs contribute only very little to the overall prediction quality, most likely due to the redundancy with other input variables (cloud data, surface albedo, sun zenith angle, etc.). The ERA5 variables "wind speed" and "vertical velocity" are vertically resolved at 1000 hPa, 950 hPa, 900 hPa, 850 hPa, 750 hPa, and 700 hPa. "Wind speed" refers to the absolute wind speed profile, i.e. $\sqrt{u^2 + v^2}$, where $u$, and $v$ are the northward and eastward wind speeds, respectively. "Boundary layer dissipation" is an ERA5 variable, which measures

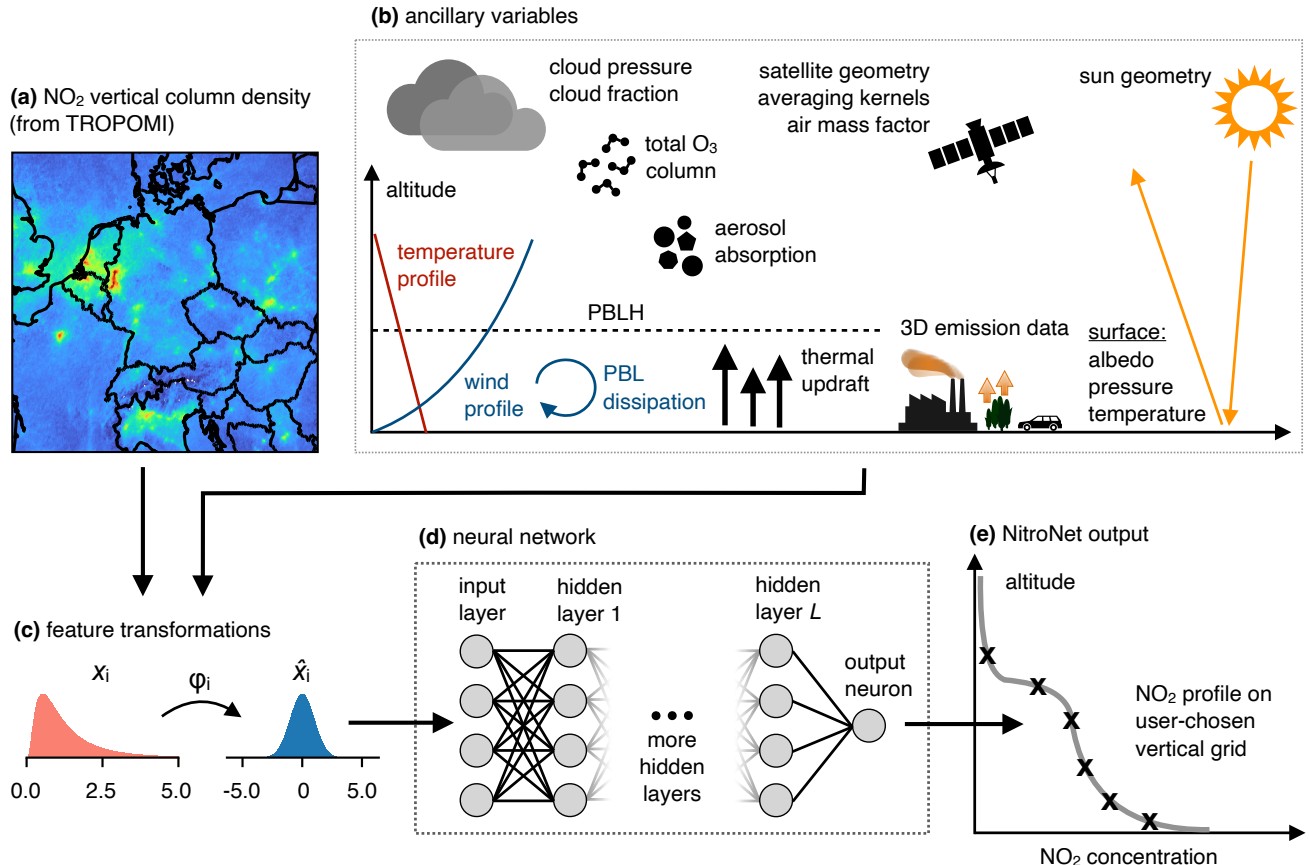

**Figure 1.** Overview of the NitroNet model. (a) and (b) depict the various input variables, which undergo feature transformation (c) before entering NitroNet's neural network (d). The output of the neural network is an $NO_2$ profile on a vertical grid of the user's choice.

the conversion of kinetic energy into heat due to small-scale eddies in the planetary boundary layer (PBL). NitroNet receives $NO_x$ emissions from the EDGARv5 emission inventory, along with the corresponding relative contribution of four emission

bins based on the "Standard Nomenclature for Air Pollution" (SNAP, see European Environment Agency (2023)). The intent is to inform the neural network about the horizontal (EDGARv5) and vertical (SNAP) distribution of emissions. The SNAP sectors used here are "1" (public power, cogeneration and district heating plants), "3" (industrial combustion), "4" (production processes) and "surface emissions", by which we refer to e.g. road traffic or agricultural emissions. NitroNet uses a ternary surface classification (urban, cropland, forest), which is available within the TROPOMI $NO_2$ product. The "VCD influx"

variable represents the amount of $NO_2$ that an observed TROPOMI pixel receives from its eight immediate neighbouring pixels due to advection. The corresponding wind speeds are taken from the ERA5 reanalysis.

An in-depth analysis of the "feature importance" of each input variable was conducted, see Fig. 2. The intention is to compute the relevance of each input variable for the model's prediction quality in a rigorous manner, here using the so-called

**Table 1.** NitroNet's input variables

| Input variable name | Data source | Note |
|---|---|---|
| NO$_2$ VCD (tropospheric) | TROPOMI | v. 2.04 |
| O$_3$ VCD (total) | TROPOMI | v. 2.04 |
| tropospheric air mass factor | TROPOMI | |
| tropospheric averaging kernels | TROPOMI | 9 lowest TM5 layers |
| cloud radiance fraction | TROPOMI | |
| cloud pressure | TROPOMI | |
| aerosol absorbing index | TROPOMI | |
| surface albedo | TROPOMI | |
| surface pressure | TROPOMI | |
| sun geometry (zenith and azimuth angle) | TROPOMI | |
| satellite viewing geometry (zenith and azimuth angle) | TROPOMI | |
| planetary boundary layer height (PBLH) | ERA5 | |
| planetary boundary layer dissipation | ERA5 | |
| surface temperature | ERA5 | |
| vertical velocity | ERA5 | see https://codes.ecmwf.int/grib/param-db/?id=135 |
| wind speed | ERA5 | total absolute wind speed, i.e. $\sqrt{u^2 + v^2}$ |
| NO$_2$ emissions (total) | EDGARv5 | |
| NO$_2$ emissions (rel. contribution from SNAP 1) | EDGARv5 | |
| NO$_2$ emissions (rel. contribution from SNAP 3) | EDGARv5 | |
| NO$_2$ emissions (rel. contribution from SNAP 4) | EDGARv5 | |
| NO$_2$ emissions (rel. contribution from surface sources) | EDGARv5 | |
| surface classification (urban / cropland / forest) | TROPOMI | ternary mask |
| day | — | binary mask ($0 =$ weekday, $1 =$ weekend) |
| VCD influx | TROPOMI + ERA5 | |
| vertical grid | — | vertical grid, on which the resulting NO$_2$ profiles are defined |

*Shapley scores* (see Štrumbelj and Kononenko (2013)). As expected, the NO$_2$ VCD is by far the most important input feature
($F = 30.9$ %), followed by the emission variables ($F = 8.9$ %) and the PBLH ($F = 6.9$ %). A detailed explanation, and further
interpretation are found in Appendix B.

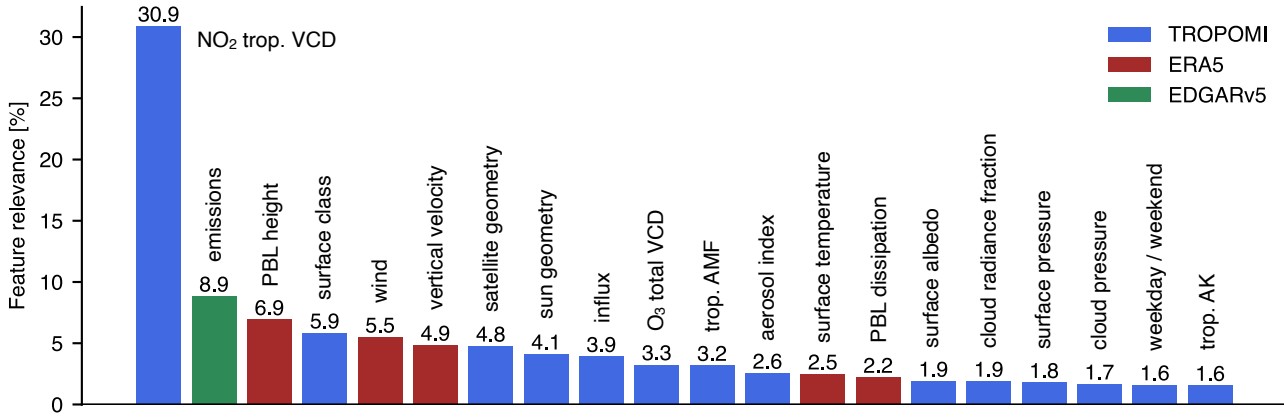

**Figure 2.** Feature relevance analysis of the NitroNet model. The legend in the top right indicates the data source of each input group.

## 3.2 Neural network design

NitroNet's neural network design is based on an extensive hyperparameter study (see Bergstra and Bengio (2012)), in which 300 different variants of the neural network (with different number of hidden layers, neurons per layer, training algorithm, etc.) were tested. The performance of a neural network can strongly depend on these parameters, but their ideal values cannot be determined on prior knowledge. The different variants were ranked based on their *mean absolute percentage error* (MAPE) on the validation set of WRF-2019. The MAPE is defined as

$$\text{MAPE}(y_{\text{pred}}, y_{\text{true}}) = \frac{1}{n} \sum_{i=1}^{N} \left| \frac{y_{\text{pred}}}{y_{\text{true}}} - 1 \right| \tag{2}$$

where $N$ is the number of instances in the validation set, $y_{\text{pred}}$ the neural network prediction, and $y_{\text{true}}$ the ground truth. The best neural network with regard to this metric was chosen for NitroNet and is described in the following.

The neural network has 8 hidden layers with 326 neurons each, corresponding to approximately 850000 trainable parameters. It uses the Parametric Rectified Linear Unit activation function (PReLU, see He et al. (2015)), the Nesterov Adam optimizer (NAdam, see Ruder (2016)), a learning rate of $3.4 \cdot 10^{-4}$, a batch size of 2048, and the $L_1$ loss function, defined as

$$L_1(y_{\text{pred}}, y_{\text{true}}) = |y_{\text{pred}} - y_{\text{true}}| \tag{3}$$

In order to reduce early stagnation of the training process as a result of too large learning rates, a simple learning rate scheduler was used (`ReduceLROnPlateau`, see Paszke et al. (2019)). The learning rate was halved whenever the training progress, as measured by the validation loss, had stalled over several epochs (meaning full iterations over the training set). Detailed information about the hyperparameter optimization procedure can be found in Appendix A. NitroNet further deploys feature

transformations (e.g. the quantile transformation from the sklearn library, see Pedregosa et al. (2012)) to reduce scale differences and skewedness of the input variables. Feature transformations are known to improve the predictive capability of machine learning models, particularly if the features or targets have a skewed or long-tailed distribution. This is the case for some of NitroNet's input features (e.g. the $NO_2$ VCD). Likewise, transformations are applied to NitroNet's training targets (the $NO_2$ concentrations at different altitudes), see e.g. Fig. C1. Prediction uncertainties are computed via the Monte Carlo method, for which a comprehensive summary is found in Anderson (1976). Figure 1 shows an overview of the NitroNet model.

## 3.3 Training NitroNet on filtered data

The overall performance of NitroNet can be significantly enhanced by the implementation of a training data filtering scheme. The idea is to rank the $NO_2$ profiles from WRF-2019 by their agreement to reference data, and only use the best few percent for training. More specifically, we define two thresholds $\Delta_{\mathrm{VCD}}$ and $\Delta_{\mathrm{PBLH}}$ and remove all training instances where

$$\left| \frac{\mathrm{VCD_{WRF}} - \mathrm{VCD_{TROPOMI}}}{\mathrm{VCD_{TROPOMI}}} \right| > \Delta_{\mathrm{VCD}} \qquad \text{or} \qquad \left| \frac{\mathrm{PBLH_{WRF}} - \mathrm{PBLH_{ERA5}}}{\mathrm{PBLH_{ERA5}}} \right| > \Delta_{\mathrm{PBLH}} \tag{4}$$

Here, $\mathrm{VCD_{WRF}}$ denotes the simulated $NO_2$ VCD from WRF-2019, $\mathrm{VCD_{TROPOMI}}$ the observed $NO_2$ VCD from TROPOMI (using the simulated $NO_2$ a priori profiles), $\mathrm{PBLH_{WRF}}$ the simulated PBLH from WRF-2019, and $\mathrm{PBLH_{ERA5}}$ the PBLH from ERA5. This way, profiles with poor agreement to the TROPOMI $NO_2$ VCD (representing the total amount of $NO_2$) or the ERA5 PBLH (representing atmospheric mixing depth and profile shape) are identified and dismissed from training. The lower $\Delta_{\mathrm{VCD}}$ and $\Delta_{\mathrm{PBLH}}$ are chosen, the fewer instances remain in the training set. Therefore, we face a trade-off between training data quality and quantity, which we resolve by including $\Delta_{\mathrm{VCD}}$ and $\Delta_{\mathrm{PBLH}}$ in the hyperparameter optimization mentioned in sect. 3.2. By this means, ideal values of $\Delta_{\mathrm{VCD}} = 0.2$ and $\Delta_{\mathrm{PBLH}} = 0.1$ were determined. With these thresholds, only the best 7 % of all profiles (approximately 100.000) remain for training. Figure C2 gives an overview of the spatial distribution of $NO_2$ VCDs after filtering, and the fraction of remaining instances across the domain.

It should be mentioned, that the TROPOMI $NO_2$ VCD and the ERA5 PBLH are quantities with significant uncertainties. For the retrieval of the tropospheric $NO_2$ VCD, the tropospheric air mass factor uncertainty (typically 20 % - 50 %) is known to dominate the overall uncertainty of the column (typically 30 % - 60 %), see e.g. Liu et al. (2021). Guo et al. (2024) report summertime ERA5 PBLH errors of approximately 150 m over continental regions, derived from radiosonde measurements. With an average PBLH of approx. 1500 m on the WRF-2019 domain, this amounts to a relative uncertainty of approx. 10 %.

However, caution is warranted: If the training dataset is manipulated in such a way, it may become unrepresentative of the "real world" (e.g. by extinction of feature modes). Evaluation on the validation set shows, that the use of filtered training data introduces a low bias of approximately $-10\%$ to the NitroNet predictions in the lower layers of the atmosphere. This bias can be determined immediately after training, stored in an altitude-dependent look-up table, and automatically subtracted from NitroNet's predictions. From a machine learning perspective, this look-up table is simply another hyperparameter, whose optimization is justified via validation on the independent test set.

### 3.4 Treatment of out-of-distribution instances

Neural networks are known to struggle when presented with out-of-distribution (OOD) instances, i.e. input data which lies outside the joint distribution of the training set. In the case of NitroNet (trained on one month of summertime RCT data in Europe), OOD instances are likely to occur in previously unseen geographical regions or seasons. The impact of OOD input variables on the neural network's performance can be detrimental, even if the neural network's sensitivity to the variable was low in the in-distribution case. In order to minimize the influence of OOD input variables, we implement a variant of the *winsorization* method (see e.g. Ruppert (2014)): First, the marginal probability density distributions $p_{x_i}(x)$ of the features $x_i$ are estimated using kernel density estimation (KDE) on the training set. Instance entries are considered OOD, if they lay in regions of relatively low probability density, e.g. if $p_{x_i}(x) < 0.15$. In that case, they are replaced with a sample from $p_{x_i}$. The $NO_2$ VCD and categorical input features (i.e. surface classifications) are exempt from this treatment. The described method is applied exclusively at prediction time. The amount of features affected depends mainly on the season and location of the input data.

### 3.5 Correction of $NO_z$ biases of in-situ measurements

An important part of the validation study presented in sect. 4 will be the comparison of NitroNet predictions to in-situ measurements at the surface. Over 90 % of the European in-situ measurements rely on the molybdenum-based chemiluminescence method, which is demonstrably cross-sensitive to other atmospheric oxidants (summarized as "$NO_z$"), such as peroxyacetyl nitrate (PAN), nitric acid ($HNO_3$) and the alkyl nitrates (see Dunlea et al. (2007); Steinbacher et al. (2007); Lamsal et al. (2008); Boersma et al. (2009); Villena et al. (2012)). Consequently, the reported $NO_2$ values are often too large, because a fraction of the $NO_z$ is falsely registered as $NO_2$. Lamsal et al. (2008) give an empiric formula for the overestimation of the $NO_2$ concentration in the presence of $NO_z$:

$$F := \frac{[NO_2^*]}{[NO_2]} = 1 + \frac{0.95[PAN] + 0.35[HNO_3] + \sum \text{alkyl nitrates}}{[NO_2]} \tag{5}$$

where [PAN], [$HNO_3$], and [$NO_2$] denote the true surface mixing ratios of PAN, $HNO_3$, and $NO_2$, while [$NO_2^*$] denotes the biased measurement result. The same formula was used in Kuhn et al. (2024), and was found to be crucial for the agreement between simulation data and in-situ measurements. NitroNet was trained to predict $F$ (as learned from WRF-2019) as an additional output, so that when comparing NitroNet predictions to in-situ measurements, the measurement bias can be compensated. Internally, this additional output is achieved by instantiating a second identical neural network, trained on the $F$ targets from WRF-2019 instead of the $NO_2$ targets. Because alkyl nitrates are not included in the MOZART chemical mechanism used in WRF-2019, we must assume $\sum \text{alkyl nitrates} = 0$. According to Elshorbany et al. (2012), the contribution of the alkyl nitrates to $F$ can be estimated in the range of 2 % - 6 %. Based on the evaluation on the test set, NitroNet can reproduce the $F$-values from WRF-2019 with a relative precision of $\pm 5$ % and no bias.

## 4 Results

### 4.1 Evaluation of NitroNet in May 2019

From hereon, we deal with the validation of the trained NitroNet model. The easiest way to confirm successful training of the model is to validate it against new examples from the test set. Figure 3a shows four exemplary $NO_2$ profiles from the test set and the corresponding predictions from NitroNet. Our model reproduces the shape and magnitude of the profiles well, although there are small deviations, e.g. in profile (C) at $\sim 3$ km altitude. Within the boundary layer, almost no discrepancies are observed. A noteworthy feature of the $NO_2$ profiles is their upper-tropospheric portion starting at 8 km altitude. Here, a sudden enhancement of the $NO_2$ concentration is found, which could be linked e.g. to aircraft emissions, decay of $NO_x$ reservoirs, lightning, or stratosphere-troposphere exchange. Figure 3b shows a scatter plot of all $NO_2$ concentrations in the (filtered) test set against their corresponding NitroNet predictions. The linear regression reveals excellent agreement, a strong correlation of $R = 0.99$, and a negligible bias of -0.4 %. The relative prediction errors are smaller at higher $NO_2$ concentrations. This is because the high $NO_2$ concentrations at the surface are more strongly correlated to the $NO_2$ VCD, which is the main model input. Vice versa, the correlation is weaker in higher layers, where the concentration tends to be lower. Therefore, the combined input variables are more descriptive of the lower, more polluted layers, and allow the neural network to make a more precise prediction. Note, that Fig. 3 shows data from the filtered test set exclusively. This choice was made for two main reasons: On one hand, we aim to exclude supposedly erroneous $NO_2$ profiles from WRF-Chem for the evaluation of NitroNet. These would result in larger errors in the comparison between WRF-Chem and NitroNet, particularly because the WRF-Chem $NO_2$ profiles show systematic errors that NitroNet does not reproduce. This is demonstrated more explicitly further below. On the other hand, the evaluation against filtered test data is an assessment of the neural network's performance in isolation, i.e. it indicates its prediction errors on instances from the same distribution as the training set. For completeness, a version of Fig. 3 based on un-filtered test data is shown in Fig. C3.

Next, we verify that the training on filtered data as described in sect. 3.3 does indeed have the desired effect. For this purpose, we inter-compare observed and simulated $NO_2$ VCDs and surface concentrations from WRF-2019, NitroNet, TROPOMI, and AirBase. Figure 4a shows the comparison of monthly-mean $NO_2$ VCDs from TROPOMI and the corresponding simulation results from WRF-2019. The simulated VCDs are computed as

$$\text{VCD}_{\text{sim}} = \sum_{l < l_{\text{tp}}} c_l \cdot \Delta h_l \tag{6}$$

where $l$ denotes the layer index, $l_{\text{tp}}$ the tropopause layer index, $c_l$ the $NO_2$ concentration in layer $l$, and $\Delta h_l$ the vertical extent of layer $l$. The $NO_2$ a priori profiles used in the air mass factor computation of the TROPOMI VCDs were replaced with those from WRF-Chem, following Eskes et al. (2019):

$$\text{VCD}_{\text{obs, corr}} = \text{VCD}_{\text{obs}} \cdot \frac{\text{AMF}_{\text{trop}}}{\text{AMF}} \cdot \frac{\sum_{l < l_{\text{tp}}} c_l \cdot \Delta h_l}{\sum_{l < l_{\text{tp}}} c_l \cdot \Delta h_l \cdot A_l} \tag{7}$$

where $\text{VCD}_{\text{obs, corr}}$ denotes the VCD with exchanged a priori profile, $\text{VCD}_{\text{obs}}$ the original VCD, AMF the total air mass factor, $\text{AMF}_{\text{trop}}$ the tropospheric air mass factor, and $A_l$ the tropospheric averaging kernel of layer $l$. Figure 4a reveals significant

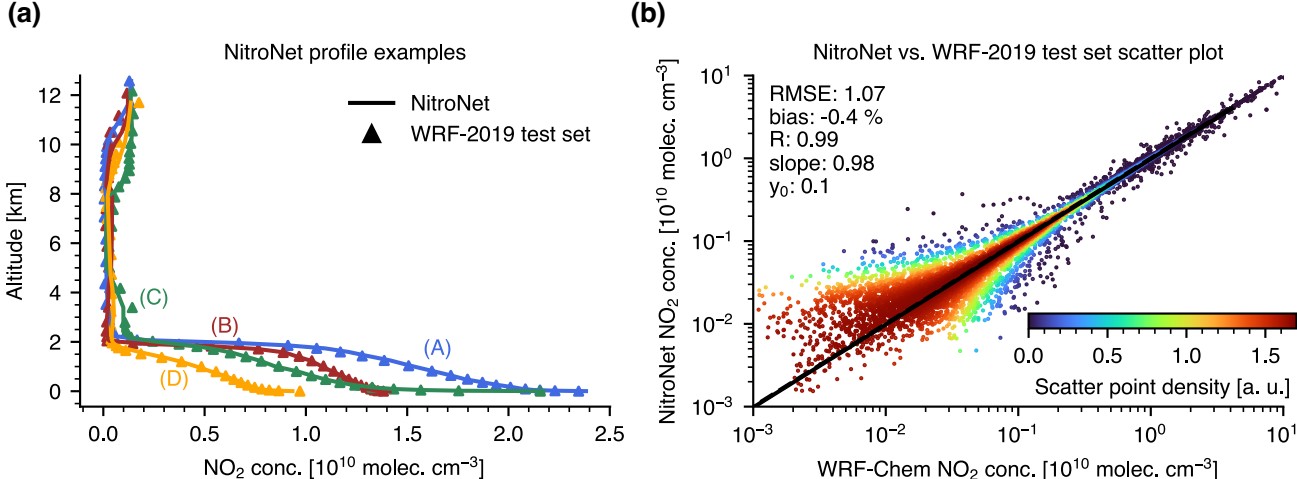

**Figure 3.** Evaluation of NitroNet on the WRF-2019 test set. **(a)** Four exemplary $NO_2$ profiles from the test set (triangular markers) with corresponding NitroNet predictions (solid lines). **(b)** Scatter plot of all $NO_2$ concentrations in the test set vs. their corresponding NitroNet predictions. RMSE and intercept are expressed in units of $10^9$ molec. $cm^{-3}$.

biases in the WRF-Chem simulation of up to $10^{16}$ molec. $cm^{-2}$ (e.g. in western Germany, northern Austria, and the Kalin-
ingrad Oblast). The simulated and observed $NO_2$ VCDs agree with a mean bias of -2.9 %, a root mean squared error (RMSE) of $6.7 \cdot 10^{14}$ molec. $cm^{-2}$, and a correlation coefficient of $R = 0.88$. Here, and throughout the rest of the article, "correlation coefficient" refers to the Pearson correlation coefficient. A more detailed discussion of the WRF-Chem simulation results can be found in Kuhn et al. (2024).

Fig. 4b shows the same comparison, but using the $NO_2$ profiles from NitroNet instead of WRF-Chem. Overall, much better
agreement is observed. In particular, the major overestimations observed with WRF-Chem have disappeared, while some weak underestimations remain. Although the absolute mean bias is slightly larger (-8.1 %), the correlation is much stronger ($R = 0.97$) and the RMSE was almost halved ($3.8 \cdot 10^{14}$ molec. $cm^{-2}$). In some regions of the domain (e.g. near the cities of Frankfurt and Mannheim, Germany), these improvements are easily explained by the considerable reduction of the simulated column. In other regions (e.g. at the border between Belgium, Netherlands, and Germany), the improvements must be partially
attributed to larger TROPOMI reference VCDs, resulting from the use of presumably more realistic a priori $NO_2$ profiles. Because the $NO_2$ VCD is the dominant input variable of NitroNet, and acts essentially as a scaling factor for the predicted $NO_2$ profiles, the relative prediction uncertainty is approximately equal to that of the $NO_2$ VCD (here: 30 % - 60 %).

Figure 5 shows the comparison of monthly-mean $NO_2$ surface concentrations from AirBase to the corresponding model results at TROPOMI overpass time. The $NO_z$ bias correction described in sect. 3.5 was applied to the AirBase data, using WRF-
2019 and NitroNet model results, respectively. The "Difference" subplots of Fig. 4 and Fig. 5 show a clear correlation, e.g. in western Germany and northern Italy. Nonetheless, different spatial patterns can be identified between NitroNet and WRF-

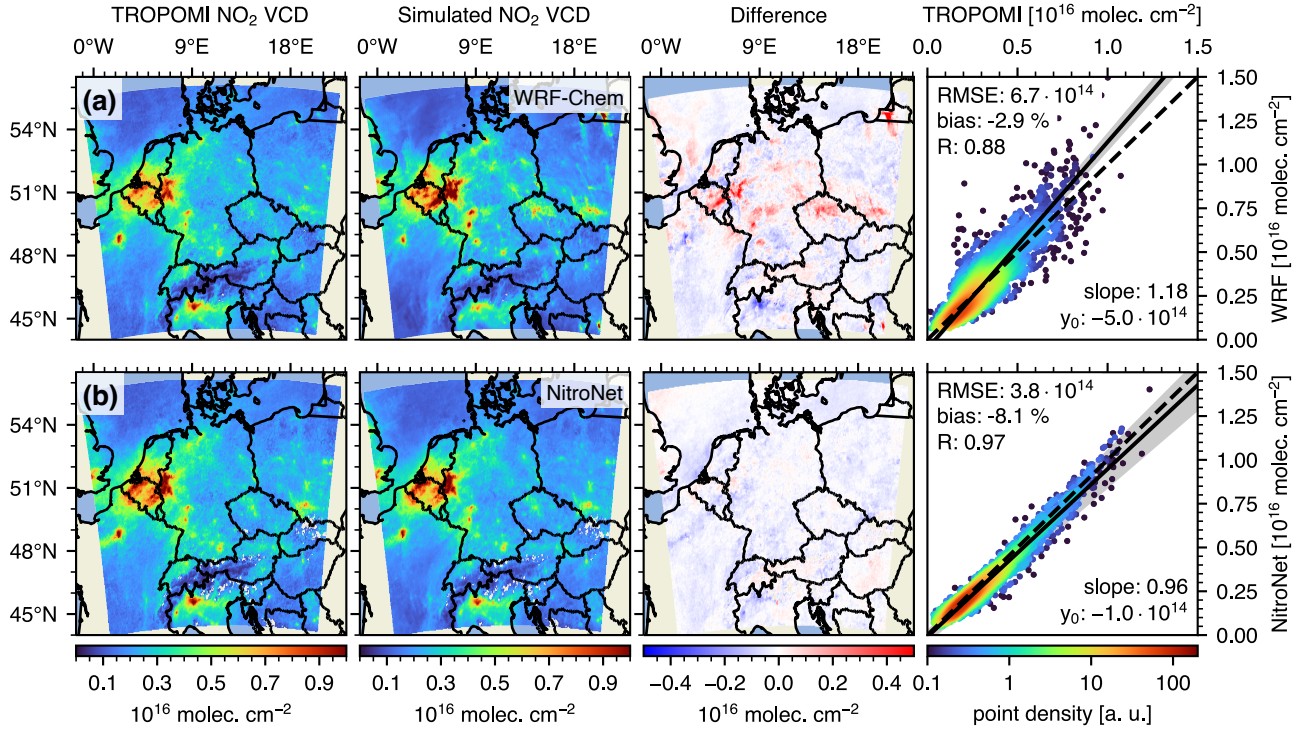

**Figure 4.** Comparison of monthly-mean TROPOMI $NO_2$ VCDs against simulated $NO_2$ VCDs from WRF-Chem **(a)** and NitroNet **(b)** (May 2019). The $NO_2$ a priori profiles used in the air mass factor computation of the TROPOMI VCDs were replaced with those from WRF-Chem and NitroNet, respectively. RMSE and intercept are given in units of $10^{14}$ molec. $cm^{-2}$.

2019: In some model regions (e.g. in western Germany) NitroNet produced smaller errors than WRF-Chem with respect to the VCDs and the surface concentrations. However, the opposite is observed in other regions. For example, NitroNet produced smaller VCD errors, but larger surface concentration errors in northern Italy. This demonstrates that filtering of the training data based on VCD and PBLH criteria alone may not always lead to better neural network predictions at the surface. Scatter plots for individual countries (Germany, Netherlands, and Italy) with differing response to the data filtering (improvement, neutral, worsening) can be found in Fig. C4 and C5. This finding is important for the interpretation of the presented results: WRF-Chem produces positive and negative errors in moderate balance, while NitroNet produces similar negative, but much smaller positive errors. Subsequently, NitroNet shows a smaller RMSE (3.2 $\mu g\ m^{-3}$ vs 3.4 $\mu g\ m^{-3}$), but larger absolute mean bias (-16.0 % vs. -11.7 %). In such a case, the increase in absolute mean bias is obviously not a suitable measure for overall model skill. The slight reduction in correlation coefficient ($R = 0.67$ vs. $R = 0.69$) escapes this argument, but can be considered insignificant.

Figure 6 shows a histogram of the $NO_z$ biases of the in-situ measurements, computed from modelled PAN and $HNO_3$ mixing ratios according to Lamsal et al. (2008), see sect. 3.5. The results obtained from WRF-Chem and NitroNet show values of up to +200 %. We show this figure with the intent of emphasizing that caution is required when using in-situ measurements

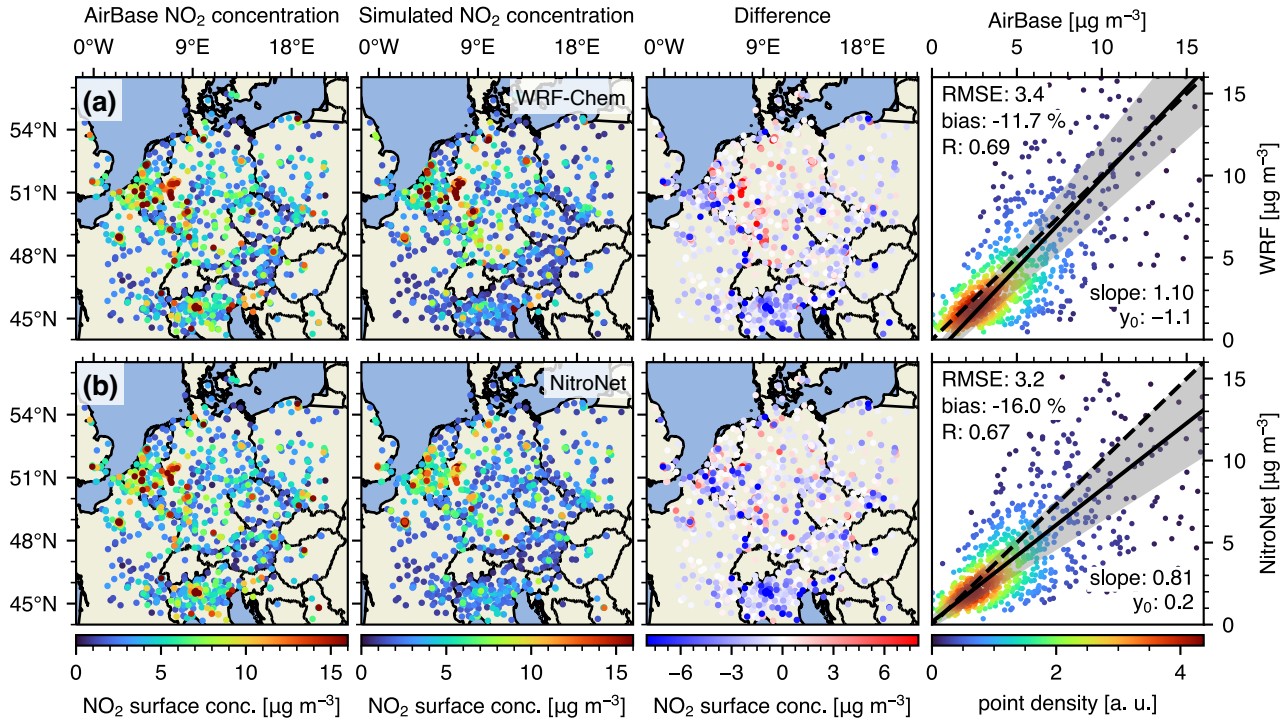

**Figure 5.** Comparison of monthly-mean AirBase $NO_2$ surface observations against simulated surface concentrations from WRF-Chem **(a)** and NitroNet **(b)** at TROPOMI overpass time (May 2019). The AirBase observations were corrected for $NO_z$ biases, using WRF-Chem model results for **(a)** and NitroNet predictions for **(b)**, respectively. RMSE and intercept are given in units of $\mu g \, m^{-3}$.

for training and validation of RCT and machine learning models without a proper correction strategy. As mentioned before, NitroNet is able to reproduce the $NO_z$ correction factors of WRF-Chem with a relative precision of $\pm 5 \, \%$ and no bias. Due to the good agreement between WRF-Chem and NitroNet in this regard, the prediction of the $NO_z$ correction factors cannot explain the low-biases observed in Fig. 5.

The results of this section demonstrate that our training method has the intended effect: Using filtered data, NitroNet produces
$NO_2$ profiles of overall more realistic magnitude and/or shape than WRF-Chem. Although the improvement to the simulated surface concentrations is rather small, a much stronger improvement to the VCDs is obtained. Even better results are expected by further filtering the training data by their agreement to the in-situ observations. However, this is impossible here, as the surface observations are so sparse that too few data would remain for the training of the neural network.

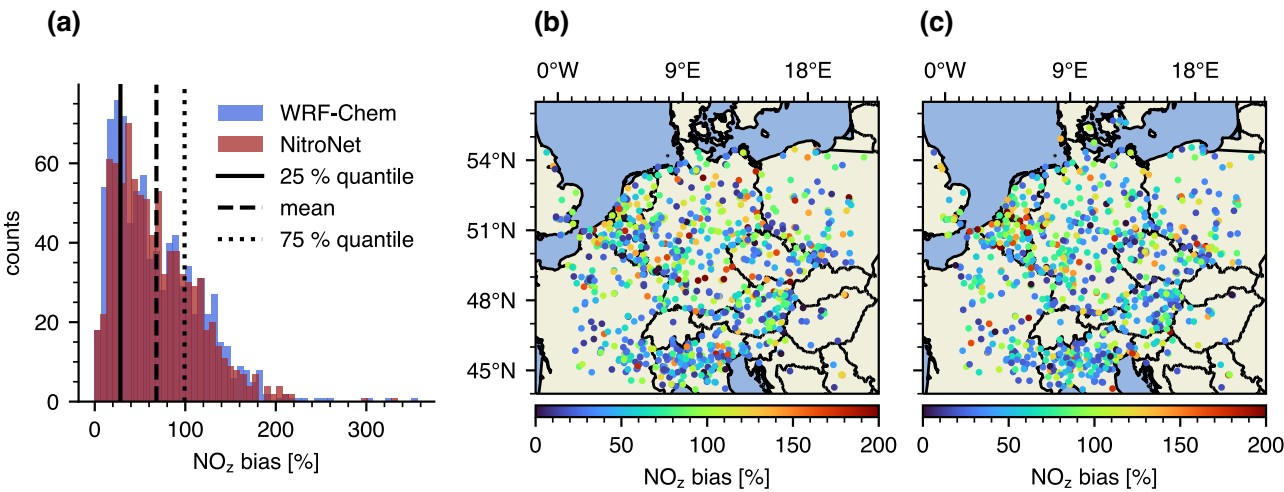

**Figure 6.** Histogram **(a)** and geographic distribution of the monthly-mean $NO_z$ biases from **(b)** WRF-Chem and **(c)** NitroNet corresponding to the AirBase observations shown in Fig. 5.

## 4.2 Evaluation of NitroNet on unseen data (May 2022)

We now address the validation of NitroNet on completely new input data from the month of May 2022. From hereon we use NitroNet without comparison to RCT simulation data on a domain ranging from 42° to 56° latitude, and from -5° to 23° longitude.

### 4.2.1 Validation against TROPOMI satellite data and AirBase in-situ measurements

Figure 7 shows the comparison of monthly-mean $NO_2$ VCDs from TROPOMI against NitroNet predictions. The computations
were conducted as explained in sect. 4.1. The NitroNet $NO_2$ VCDs show similar magnitudes, geographical distribution, and errors as in May 2019. However, the results for May 2022 show lower RMSE ($2.8 \cdot 10^{14}$ molec. cm$^{-2}$ vs. $3.8 \cdot 10^{14}$ molec. cm$^{-2}$), and increased mean bias (+6.7 % vs. -8.1 %). This apparent improvement could be purely coincidental: Figure 4b indicates a slight underestimation of the $NO_2$ VCDs on behalf of NitroNet. On the other hand, the $NO_2$ VCDs in May 2019 were on average 18 % higher than in May 2022; Subsequently, NitroNet can overestimate the true VCDs because it attempts to repro-
duce the approximate magnitudes learned from 2019. If the two effects cancel each other out, this could reasonably explain the smaller VCD errors observed in 2022.

Figure 7b shows the comparison of monthly-mean $NO_2$ surface concentrations from AirBase against NitroNet predictions. NitroNet correctly identifies surface pollution hotspots (e.g. in Paris (France), Essen (Germany), and Hamburg (Germany)), but somewhat underestimates surface $NO_2$ concentrations in various regions of the domain. Compared to May 2019, the results
show a smaller mean bias (-10.5 % vs. -16.0 %), a higher correlation coefficient ($R = 0.72$ vs. $R = 0.67$), and significantly

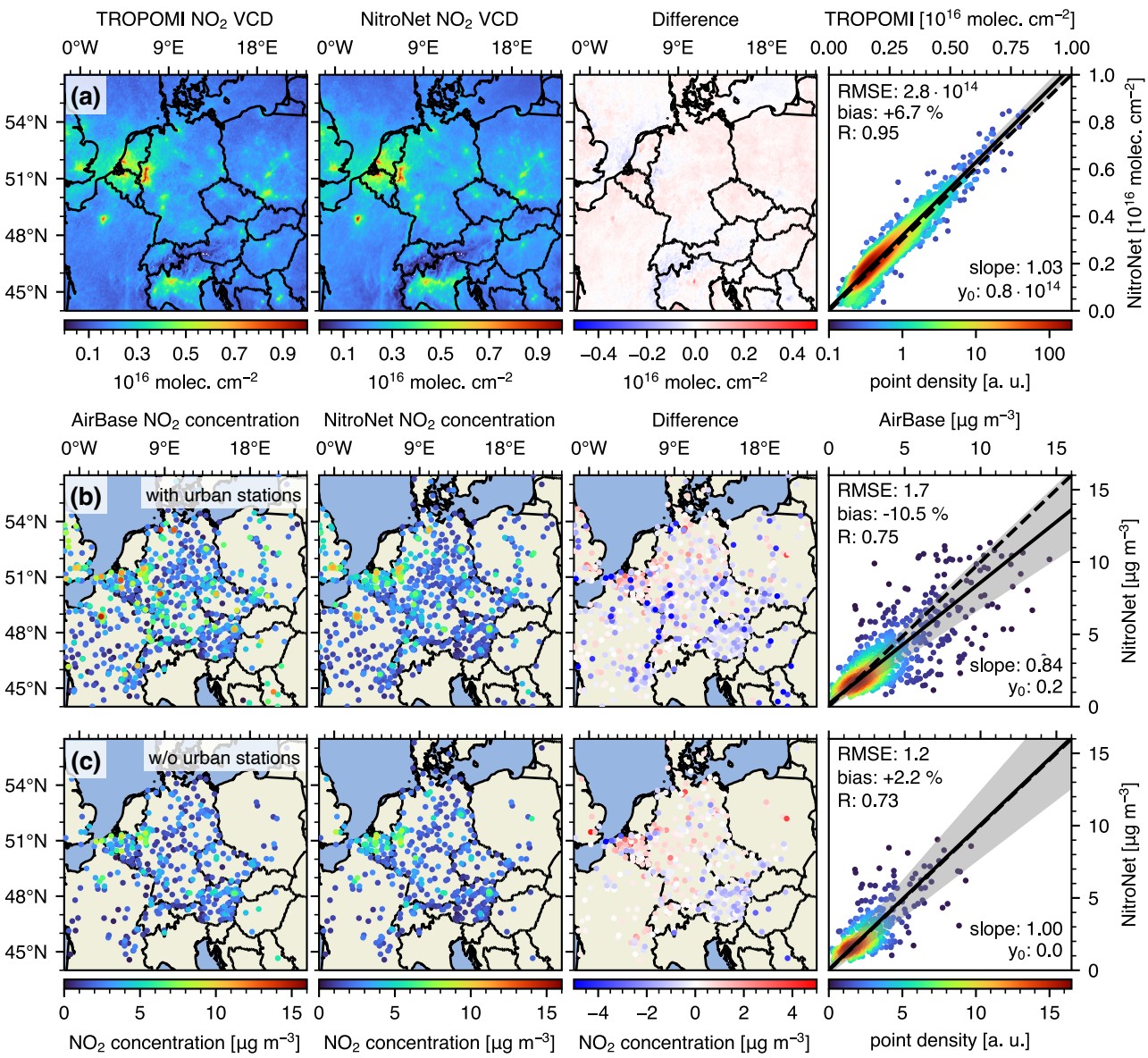

**Figure 7.** Comparison of monthly-mean TROPOMI $NO_2$ VCDs **(a)** and AirBase surface observations **(b)** against NitroNet predictions (May 2022). Subfigure **(c)** is identical to **(b)**, except that AirBase instruments of the type "urban background" were removed. RMSE and intercept are displayed in molec. $cm^{-2}$ for the VCDs, and $\mu g\ m^{-3}$ for the surface concentrations.

reduced RMSE (1.2 $\mu$g m$^{-3}$ vs 3.2 $\mu$g m$^{-3}$). A key contribution to these differences is found in the Lombardy region of northern Italy. Here, significant underestimations were observed in 2019, but the corresponding data points are missing entirely in 2022. Inspection of the AirBase metadata reveals, that in May 2019 over 92 % of the Italian measurements were flagged as "valid", 5 % as "invalid", and 2 % as "below detection limit". In May 2022, however, only 48 % of the measurements were flagged as "valid", 13 % as "invalid", and 39 % as "below detection limit". Additionally, the total number of Italian instruments was reduced from 320 in 2019 to just 69 in 2022. It remains unclear, why these measurements were removed from AirBase.

Another interesting observation is the dependence of NitroNet's low bias on the measuring stations' type. Here we refer to the entire domain shown in Fig. 7. As explained in sect. 2.2, we exclusively use background stations throughout our study, based on the argument that accurate modelling of traffic and industrial scenarios is known to require simulations of much higher resolution ("local scale"). So far we have assumed no errors in the classification of the AirBase instruments. However, based on the resolutions of modern emission inventories, the variability of trace gas transport, and the scarce documentation on classification criteria, it can be argued that the category "urban background" is a grey zone within this classification. After all, emission inventories clearly show that urban regions are always affected by traffic emissions. Furthermore, Fig. 7 shows significant low biases in NitroNet's surface predictions, but no corresponding low biases in the tropospheric columns. This can partly be attributed to the inter-pixel variability of the TROPOMI measurements. Surface stations with a large NitroNet bias are possibly located closer to strong traffic emissions, and thus less correlated with the NO$_2$ VCD, which acts as the main model input. We therefore investigated, whether the comparison of NitroNet's results to in situ observations would improve by removing the urban background stations, as shown in Fig. 7c. Significant improvements were revealed, manifesting in increased slopes (from 0.84 to 1.00), lower absolute mean bias (-10.5 % to +2.2 %), and lower RMSE (1.7 $\mu$g m$^{-3}$ to 1.2 $\mu$g m$^{-3}$). These improvements can be explained either by a tendency of NitroNet to underestimate NO$_2$ concentrations in urban areas, or by an ambiguous categorization of the measurements. Due to the lack of information about the classification process, we will omit the urban background stations in our evaluations from hereon.

### 4.2.2 Validation against FRM$_4$DOAS MAX-DOAS measurements

We now validate the NO$_2$ profiles from NitroNet against MAX-DOAS measurements from the FRM$_4$DOAS dataset at six European locations. A temporal threshold of 60 minutes is used, meaning that each NitroNet NO$_2$ profile is associated with the average over all colocated MAX-DOAS profiles recorded within 60 minutes of the corresponding satellite overpass. Averaging kernels are available from the MMF retrieval algorithm and given as an $n \times n$ matrix $\mathbf{A}$, where $n$ denotes the number of vertical layers in the retrieval. The $i$-th row of $\mathbf{A}$ describes the retrieval sensitivity of the concentration value of layer $i$ to the other $n$ layers. An ideal retrieval would be characterized by $\mathbf{A} = \mathbb{1}$, where $\mathbb{1}$ denotes the unity matrix. In practice, the AK matrix diagonal is usually close to unity at the surface, but quickly drops below 50 % within the first 1-2 km above ground (see e.g. Fig. C7, showing the AK matrix of the instrument in Heidelberg, Germany). The AKs are applied to the NitroNet profiles following Rodgers (2000) by computing

$$c_{\text{sim, corr}} = \mathbf{A}c_{\text{sim}} + (\mathbb{1} - \mathbf{A})c_{\text{ap}} \tag{8}$$

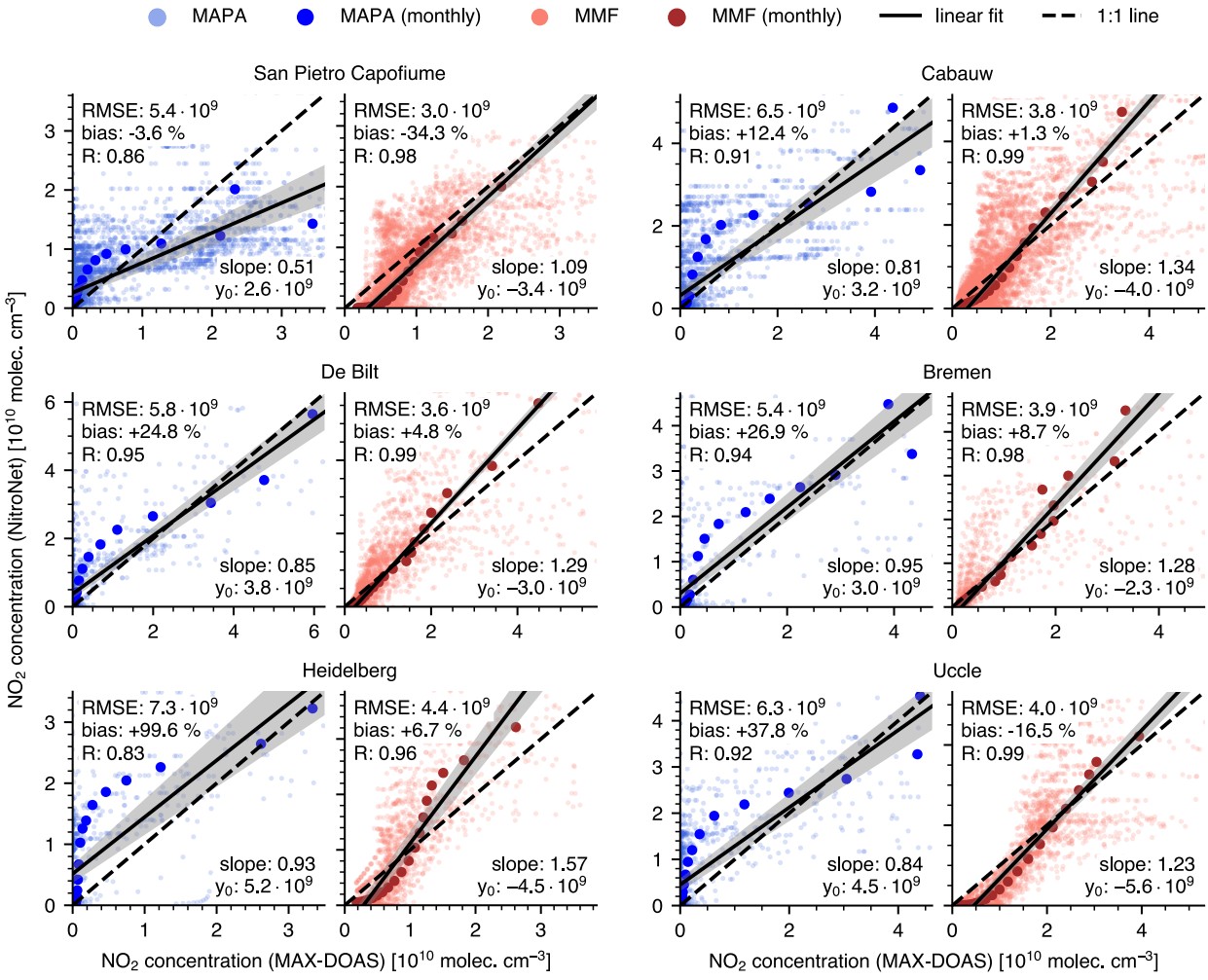

**Figure 8.** Comparison of FRM$_4$DOAS NO$_2$ concentrations against NitroNet predictions (May 2022). MAPA results are drawn in blue, and MMF results in red. The thin scatter points represent a one-to-one comparison of NO$_2$ concentration values (i.e. the concentrations of individual profiles). The thick scatter points show the monthly-mean NO$_2$ concentrations of each retrieval layer. RMSE and intercept are displayed in molec. cm$^{-3}$ and were computed based on the monthly-mean scatter points.

where $c_{\mathrm{sim}}$ denotes the original NitroNet profile and $c_{\mathrm{ap}}$ the assumed a priori profile. The AKs are applied as described when comparing NitroNet to MMF profiles. MAPA, on the other hand, does not supply AKs.

Figure 8 shows the results obtained with this procedure. The thin scatter points (legend handles "MAPA" and "MMF") represent a one-to-one comparison of NO$_2$ concentration values from NitroNet and MAPA or MMF. The thick scatter points (legend handles "MAPA (monthly)" and "MMF (monthly)") show the monthly-mean NO$_2$ concentrations of each retrieval layer. The level of agreement between FRM$_4$DOAS and NitroNet varies, depending on the instrument location. NitroNet and

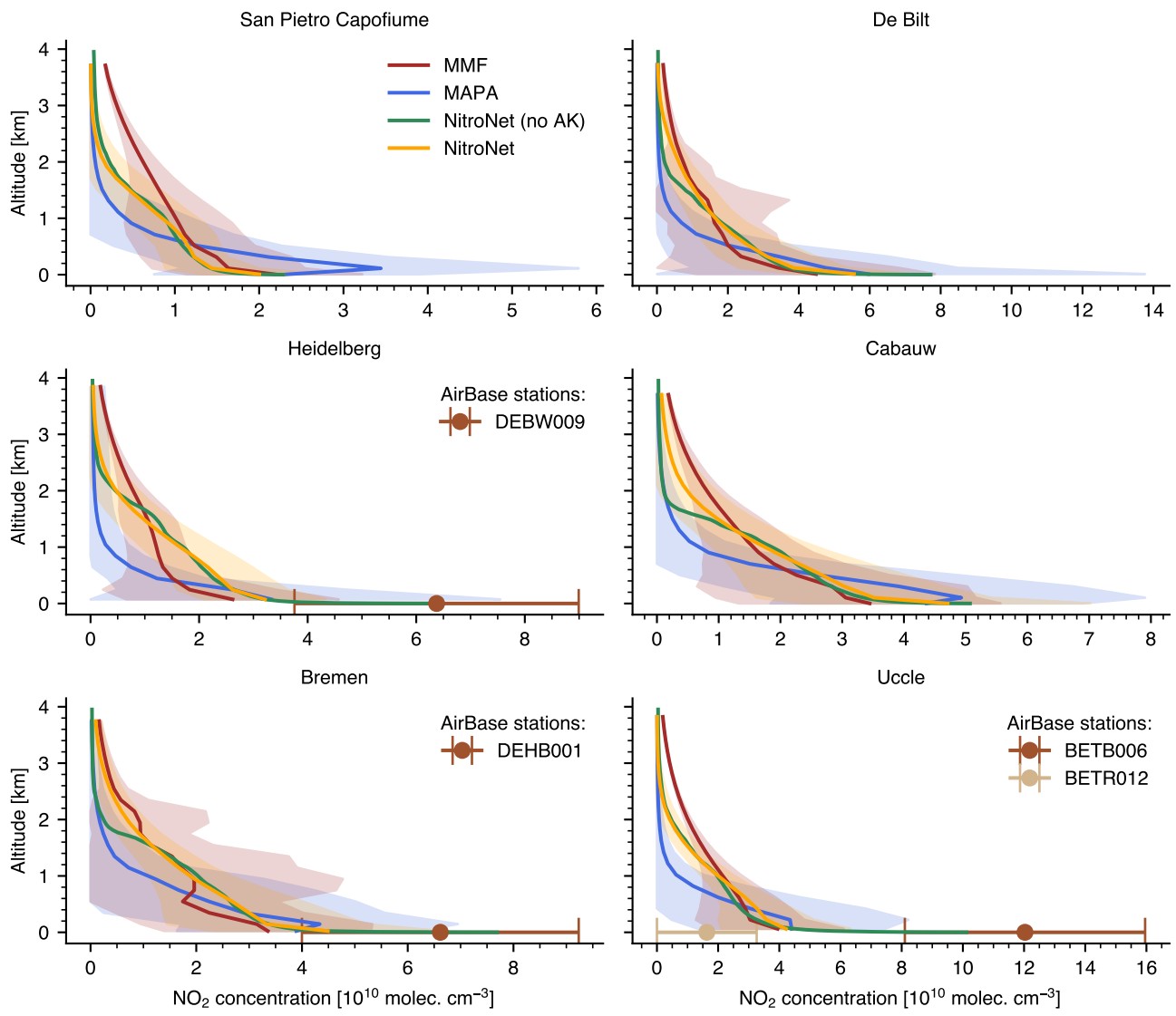

**Figure 9.** Comparison of monthly-mean FRM$_4$DOAS NO$_2$ profiles against NitroNet profiles (May 2022). The monthly standard deviations of the profiles are drawn as shaded regions in the background. Where available, colocated AirBase measurements of the surface NO$_2$ concentration within a radius of 5 km were drawn at 0 m altitude.

MAPA show significant differences in some locations, with biases ranging from $-3.6\%$ (San Pietro Capofiume) to $+99.6\%$ (Heidelberg), RMSE values on the scale of $6 \cdot 10^9$ molec. cm$^{-3}$ and correlation coefficients ranging from $R = 0.86$ (San Pietro Capofiume) to $R = 0.95$ (De Bilt). NitroNet and MMF show overall better agreement, with biases ranging from $-34.3\%$ (San Pietro Capofiume) to $+8.7\%$ (Bremen), RMSE values on the scale of $4 \cdot 10^9$ molec. cm$^{-3}$ and correlation coefficients larger than 0.90. The linear regressions show significantly steeper slopes for MMF than for MAPA, but similar intercepts.

MAPA tends to produce higher $NO_2$ concentrations than MMF in the lowest few hundred meters above ground, but smaller concentrations above. The NitroNet predictions are somewhere in between, manifesting in an "S"-shaped distribution of the scatter markers (see e.g. the comparison to MMF in Heidelberg). The corresponding plots of monthly-mean $NO_2$ profiles can be found in Fig. 9. Additionally, colocated measurements from in situ measurements (within a radius of 5 km) were drawn in the corresponding subplots of Fig. 9. NitroNet shows good agreement with the surface observations (except for the station

"BETR012" in Uccle). This is made possible by NitroNet's high vertical resolution at the surface ($\sim 1$ m), which is adequate for the steep prevailing concentration gradients. This is not the case for MAPA and MMF, because the vertical sampling of FRM$_4$DOAS ($\sim 200$ m) is too coarse. Our observations in this regard align well with the findings of Bösch (2018), who presents a a detailed comparison of MAX-DOAS measurements and colocated surface observations. The differences between MAPA, MMF, and NitroNet can partly be linked to the models' implementations and limitations: MMF uses a single, fixed

$NO_2$ a priori profile for all retrievals, which was obtained from a WRF-Chem simulation in Mexico (Friedrich et al. (2019)). However, datasets like our own WRF-2019 show strong horizontal variability of $NO_2$ profiles on the scale of just a few kilometers. A single a priori profile is therefore not sufficient to fully represent the diversity of profile shapes and magnitudes. Moreover, horizontal gradients also systematically affect the MAX-DOAS profile retrievals. Subsequently, it is not surprising to see larger differences between MMF, MAPA and NitroNet (without AKs) in the regions of reduced sensitivity (small AKs)

above 1 km altitude. Application of the AKs reduces the differences significantly in 3/6 locations (De Bilt, Cabauw, Bremen). MAPA, on the other hand, makes a priori assumptions in the form of a pre-defined profile parametrization. The profiles shown in Fig. 9 are qualitatively similar to those from MAPA's original publication paper (Beirle et al. (2019)), with a strong exponential shape and an optional peak in the 2nd or 3rd layer (San Pietro Capofiume and Cabauw). This could indicate the presence of an elevated $NO_2$ layer. NitroNet is unable to reproduce this profile type, most likely because the training dataset contains very

few corresponding examples. As shown in Kuhn et al. (2024), the WRF-Chem model, which provides NitroNet's training data, also struggles to reproduce elevated layers in some locations. On the other hand, the elevated layers are not reproduced by MMF either. In that regard, it is possible that they are falsefully produced by an incompatibility between the true $NO_2$ profile and MAPA's profile parametrization (technically a form of "model misspecification error"). Overall, the differences between MAPA and MMF demonstrate the large uncertainty from the choice of retrieval algorithm alone. Further sources of uncertainty

(e.g. the influence of horizontal gradients), as well as the low statistical relevance of only six measurement locations must be considered withal, and are not easily quantifiable. Within these limitations, the comparison to MAX-DOAS data shows no glaring discrepancies, as much as it allows for no more than an approximate validation of profile shapes and magnitudes.

## 4.3 Evaluation of NitroNet in other seasons and regions of the world

Lastly, we present an analysis of NitroNet's ability to generalize to other seasons and regions of the world. The evaluations shown in sect. 4.2.1 were made for the same region (central Europe) and time of the year (May), on which the neural network was trained. Hence they represent the least challenging test case. Good generalization to other domains and seasons is not guaranteed, and associated with two challenges: Firstly, the neural network must respond reasonably to fundamentally different input data (e.g. much lower temperatures in winter than in summer). This is controlled by the network's regularization, which we enforce mainly via the winsorization technique described in sect. 3.4. Secondly, the training data is expected to be "epistemically incomplete", meaning that it does not contain all relevant training examples for other seasons and regions. This is a property of the training set, which we regard as a principal limitation that cannot be resolved in the scope of this article. Nonetheless it is not implausible, that the fundamental relationships between the input and output data, as learned by NitroNet, hold at least partly for other seasons and regions, as well.

We first investigate the regional generalization capability of the model using reference data from May 2022. Figure 10 shows the comparison to TROPOMI $NO_2$ VCDs over the United Kingdom (UK, Fig. 10a) and the Mediterranean region of Portugal and Spain (Fig. 10b). The results are overall very similar to those from the central European domain investigated previously. However, Fig. 10b shows significant overestimations of approximately $10^{15}$ molec. $cm^{-2}$ over the southern waterbodies (the Alboran Sea and the Gulf of Cadiz). It is not generally unexpected to see such systematic errors in the predictions of a neural network. The most likely explanation is that the training dataset does not contain enough representative examples of $NO_2$ profiles over water. The water regions of the training set must be assumed less representative, e.g. because they are pervaded by unusually many shipping routes, which may lead NitroNet to overestimate $NO_2$ over more remote water bodies. We exclude these pixels from the statistical analysis, because they skew the results over the landmasses, over which we aim to validate NitroNet in this article. Compared to the central European domain, the RMSE values are increased from $2.8 \cdot 10^{14}$ molec. $cm^{-2}$ to $3.3 \cdot 10^{14}$ molec. $cm^{-2}$ (UK) and $3.1 \cdot 10^{14}$ molec. $cm^{-2}$ (Spain and Portugal), while the correlation coefficients are reduced from $R = 0.95$ to $R = 0.92$ (UK) and $R = 0.86$ (Spain and Portugal). The mean biases are +12.3 % (UK) and +3.4 % (Spain and Portugal), respectively. For context, an RMSE of $5.0 \cdot 10^{14}$ molec. $cm^{-2}$, a bias of +18.0 %, and a correlation coefficient of $R = 0.74$ is obtained for the domain of Spain and Portugal if water pixels are included. The statistical analysis of the UK domain, however, is practically unaffected by water pixels. Figure 11 shows the corresponding comparison to AirBase surface observations in analogy to Fig. 7, including the omission of "urban background" stations. A version of Fig. 11 including urban stations is found in Fig. C6. The results are similar: On the UK domain, the RMSE is slightly increased from 1.2 $\mu$g $m^{-3}$ to 1.8 $\mu$g $m^{-3}$, and the correlation coefficient is reduced significantly from $R = 0.73$ to $R = 0.45$. This is caused by the two outliers in the south-eastern corner of the domain and amplified by the low number of total observations. On the Mediterranean domain the number of observations is much larger, and the results are overall better, with an RMSE of 1.6 $\mu$g $m^{-3}$ and a correlation coefficient of $R = 0.71$. This demonstrates, that NitroNet can generalize to new, but qualitatively similar domains with minor loss of prediction accuracy. NitroNet was also tested on three more "distant" domains covering the United States (US) west coast, India, and western China (see Fig. 12). We obtain good agreement for the US west coast (RMSE =

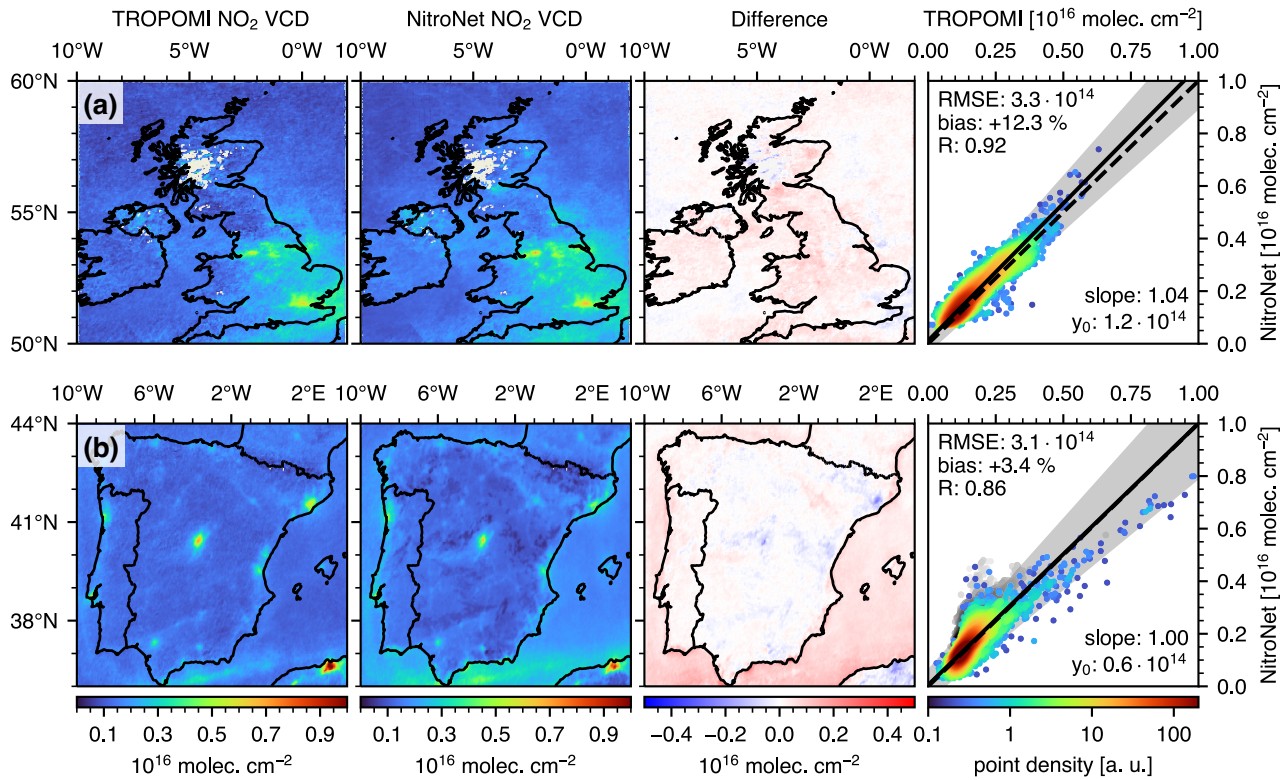

**Figure 10.** Like Fig. 7a, but for **(a)** the UK, and **(b)** the countries of Spain and Portugal. Water-pixels are drawn as gray dots in the right-side scatter plots and excluded from the statistical analysis. RMSE and intercept are displayed in molec. cm$^{-2}$.

$2.7 \cdot 10^{14}$ molec. cm$^{-2}$, bias = +2.7 %, R = 0.84). The Indian domain shows stronger correlation, but lower accuracy due to significant overestimations (RMSE = $8.0 \cdot 10^{14}$ molec. cm$^{-2}$, bias = +41.5 %, $R = 0.91$). The biggest deviations and weakest correlations are observed on the Chinese domain (RMSE = $12.6 \cdot 10^{14}$ molec. cm$^{-2}$, bias = +12.5 %, $R = 0.70$). Here, as

shown in Fig. 12c, NitroNet ignores entire pollution hotspot areas in the northern Shanxi and Shaanxi provinces. These regions are known for their strong emissions from coal, steel, chemical and military industry (see e.g. Peng et al. (2023)). China's rapid economic development combined with fewer environmental state regulations make it plausible, that the EDGARv5 emission data of the year 2015 might already be outdated in such locations. Besides, NitroNet may struggle with the differences in atmospheric composition, e.g. the vastly higher aerosol pollution which prevails in China (see e.g. Meng et al. (2022)). The

previously mentioned overestimation over waterbodies is observed in all three domains.

Finally, we investigate the seasonal performance of NitroNet. For this purpose, a whole year of data (August 2021 - July 2022) was processed on the central European domain. The NitroNet predictions were evaluated against TROPOMI and AirBase observations, and time series of the bias, RMSE, and correlation coefficient were computed, see Fig. 13. Shown here are daily mean values, as well as monthly mean values in analogy to the other evaluations presented up to this point. Note, that in

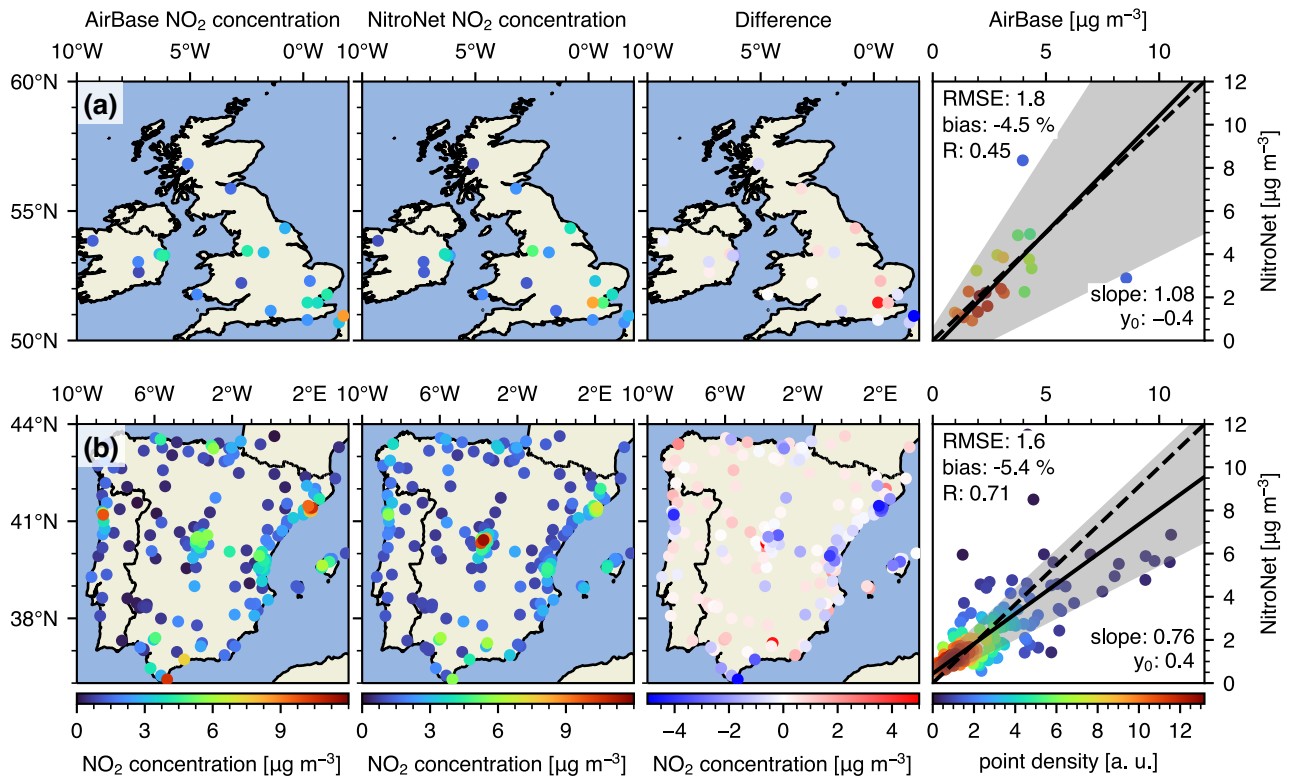

**Figure 11.** Like Fig. 7c, but for (**a**) the UK, and (**b**) the countries of Spain and Portugal. RMSE and intercept are displayed in $\mu g\, m^{-3}$.

this context "monthly-mean bias" refers to the bias computed on monthly means, as opposed to the monthly mean of daily biases (which can be estimated from the daily values shown in Fig. 13). The same holds for the RMSE and the correlation coefficient. Because averaging over multiple days reduces the noisiness of the NitroNet predictions, the monthly-mean RMSE values are smaller, and the correlation coefficients larger, than on unaveraged data. The mean biases, however, are unaffected by averaging. In the following, we will focus on the monthly means. NitroNet's performance shows a clear seasonal cycle: The mean biases increase during wintertime and reach maximal values of -22.4 % (vs. TROPOMI, January) and -50.1 % (vs. AirBase, December). Likewise, the RMSE increases during wintertime and reaches maximal values of $10.8 \cdot 10^{14}$ molec. $cm^{-2}$ (vs. TROPOMI, January) and 6.3 $\mu g\, m^{-3}$ (vs. AirBase, December). The correlation coefficients are on the scale of $R \approx 0.90$ vs. TROPOMI and $R \approx 0.70$ vs. AirBase, with no conclusive annual cycle. The decrease in model performance in winter is expected due to the reasons discussed earlier. In particular, the oxidative capacity (via the hydroxyl and peroxy radicals) is reduced in winter and results in increased $NO_2$ lifetimes of more than 20 hours, as opposed to 2 - 6 hours in summer (see e.g. Liu et al. (2016); Shah et al. (2020)). The results show that without specifically training on wintertime data, NitroNet's prediction for deep winter are only of limited value. Besides the obvious challenge of achieving good generalization from summertime training data to wintertime predictions, higher uncertainties in the input satellite data should also be taken into

account in this context (see e.g. Douros et al. (2023)). Nonetheless, compared to the typical performance of RCT simulations, NitroNet performs well for the majority of the analyzed time series. Compared to WRF-2019, with equivalent filter criteria, the RMSE values of NitroNet's $NO_2$ VCDs and surface concentrations are lower in 9 out of 12 months. Likewise, the correlation coefficients are larger in 10 out of 12 months (see the dotted gray lines in Fig. 13). It should be noted, that the performance of RCT simulations is expected to also drop significantly in wintertime. The scientific literature on the topic is sparse, but a study by Douros et al. (2023) shows that CAMS (an ensemble model consisting of 11 RCT models) produces summertime VCD biases of $\sim 15\%$ and wintertime VCD biases of $\sim 50\%$ in Europe. In light of such results, NitroNet's seasonal performance on the European domain can be considered competitive to most recent RCT simulations. Figure C8 shows examples of the comparison between NitroNet and TROPOMI for two individual days in summer and winter. In contrast to the monthly-mean comparisons shown previously, the data contains a significant amount of gaps (e.g. due to clouds), the correlation is reduced ($R \approx 0.80$), and the prediction errors are larger. This is expected, since averaging over an entire month of data reduces the statistical noise of the model. Nonetheless, as reflected in Fig. 13, NitroNet's daily performance is still competitive to that of WRF-Chem, indicating that it can reasonably be used for unaveraged predictions. A version of Fig. 13 with urban stations included is found in Fig. C9.

Figure 14 shows a full-year evaluation of NitroNet against $NO_2$ concentrations from $FRM_4DOAS$ in selected altitude ranges. For this analysis, NitroNet's average bias (left panel) and absolute error (right panel) over all previously shown FRM4DOAS instruments were computed for a full year of data, with either MMF or MAPA used as reference. Each subplot of Fig. 14 is restricted to a specific altitude range (0 - 200 m, 200 - 400 m, 400 - 600 m, 600 - 1000 m, 1000 - 2000 m). In the lowest evaluation layer, at 0 - 200 m, there is particularly good agreement between MAPA and MMF, with NitroNet biases between $-70\%$ and $+20\%$ over the course of the year. Here, a similar tendency as in Fig. 13 can be observed, with low biases occurring during winter, and high biases during summer. The summertime high biases are of similar magnitude than in the comparison to TROPOMI VCDs and AirBase surface measurements (approximately $+15\%$ vs. $+23\%$, and $+10\%$, respectively). Particularly in the higher layers, the validation against MMF yields far lower mean biases, mostly in the range from $-30\%$ to $+30\%$, while the validation against MAPA result in larger biases of   100 % at 600 - 1000 m, and   200 % at 1000 - 2000 m. This owes to the steeper vertical concentration gradients of the MAPA profiles due to their assumed profile shape, and aligns well with the profiles shown in Fig. 9. The large relative biases of NitroNet in relation to MAPA might appear concerning at first, and should be put into perspective based on the following considerations:

First, it is hard to assess, which of the two retrieval algorithms yields more trustworthy results. Although conceptionally different, MAPA and MMF both suffer from increasingly poor sensitivity at higher altitudes. This is also the case here, as exemplified by the MMF averaging kernels shown in Fig. C7, which indicate an effective vertical sensitivity of up to 1.5 km in Heidelberg, May 2022. In consequence, the retrieval results are considerably affected by a priori assumptions. In the case of MMF, an a priori profile is taken from a WRF-Chem simulation over Mexico (see Friedrich et al. (2019)), which might be entirely unrepresentative of the central European domain investigated here. Parametrized retrievals such as MAPA do not require a priori profiles, which is an advantage in this context. Nonetheless, MAPA still depends on other a priori assumptions,

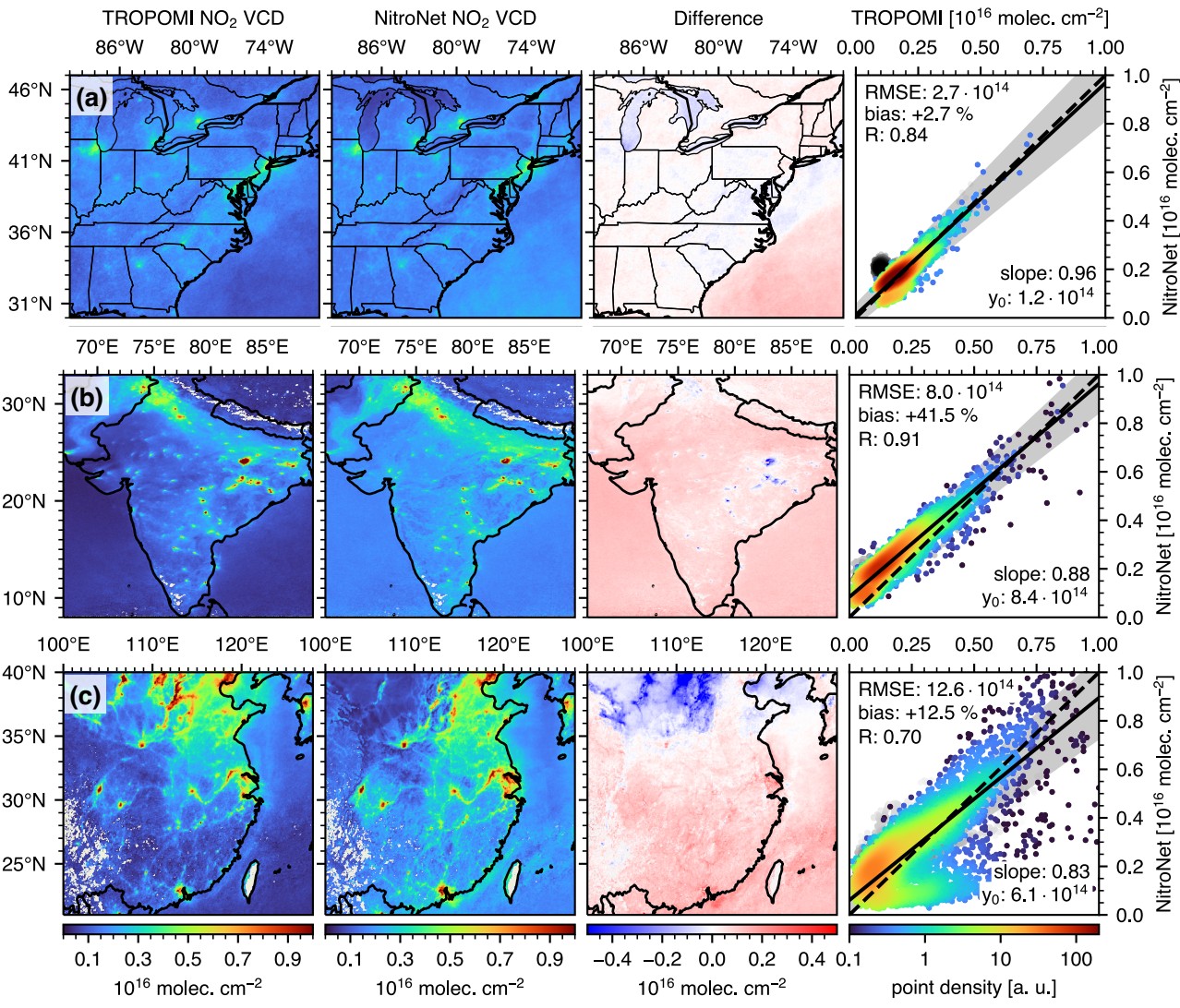

**Figure 12.** Like Fig. 10, extended to **(a)** the US west coast, **(b)** India, and **(c)** western China. The uncoloured scatter markers in panel **(a)** symbolize the entries over water, which were dismissed from the statistical analysis.

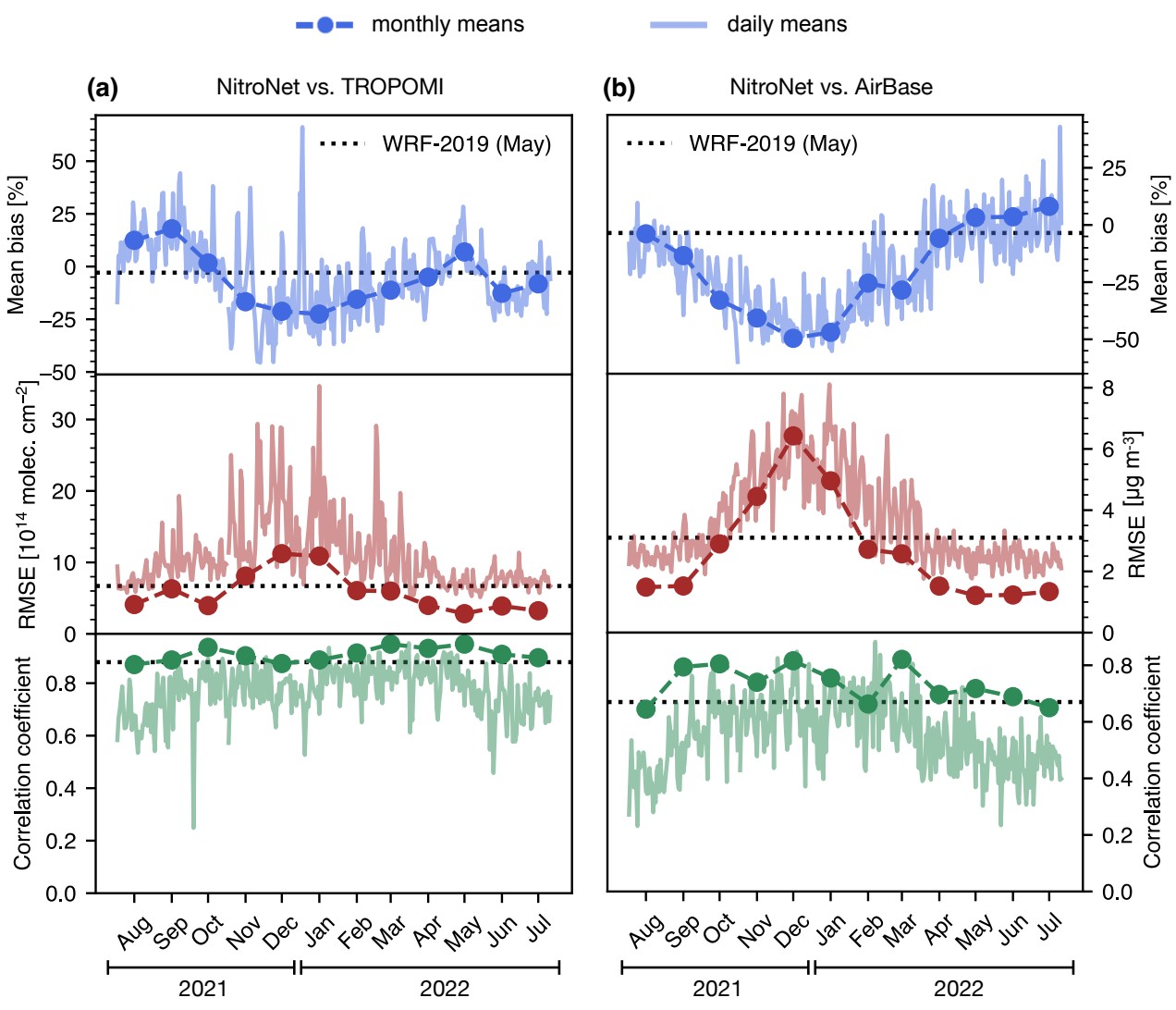

**Figure 13.** Seasonal evaluation of NitroNet on the central European domain against **(a)** NO$_2$ VCDs from TROPOMI and **(b)** surface observations from AirBase. The gray dotted line "WRF-2019 (May)" shows the value of the statistical diagnostics (mean bias, RMSE, and correlation coefficient) obtained from WRF-2019 for comparison.

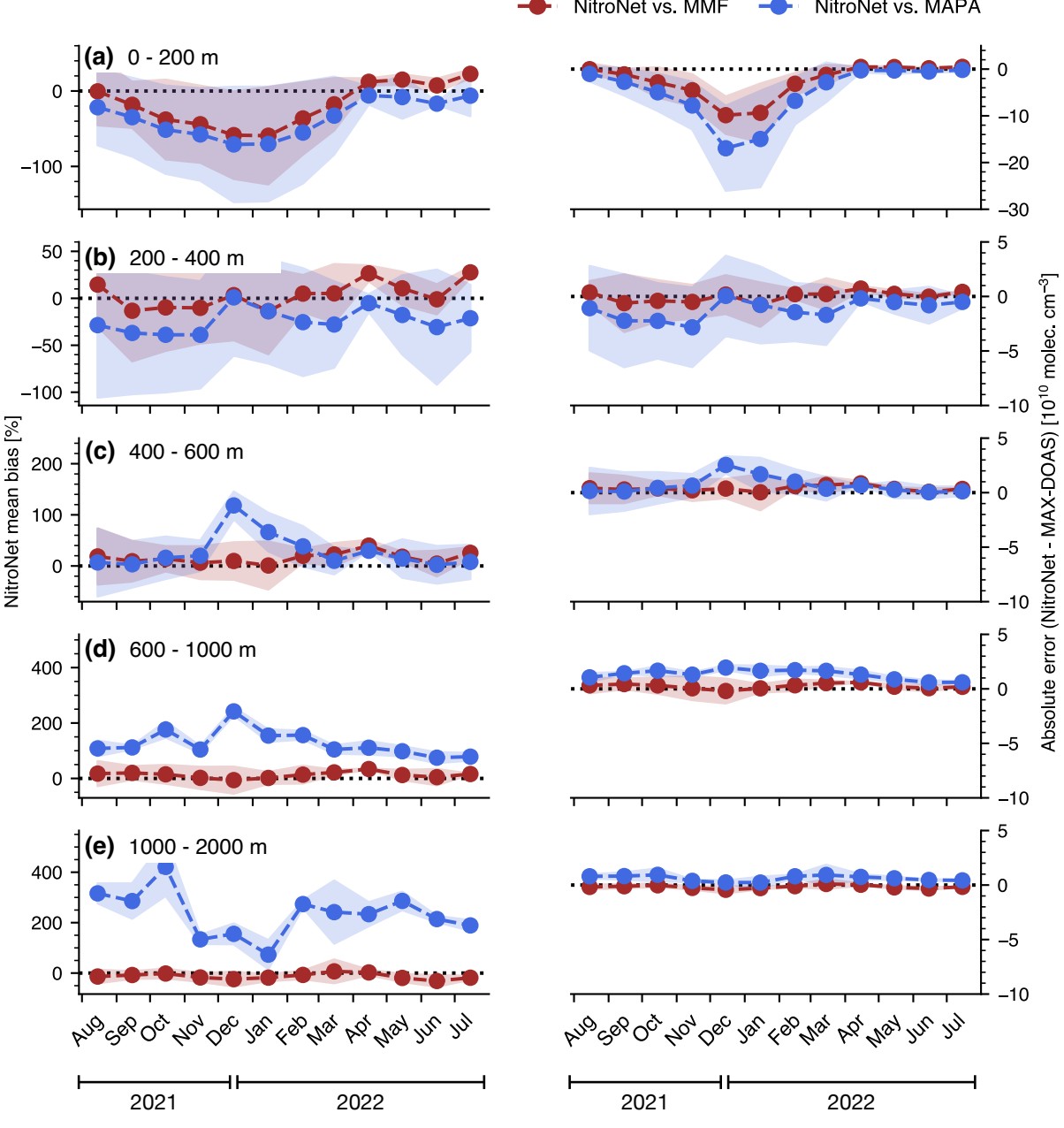

**Figure 14.** Seasonal evaluation of NitroNet against $NO_2$ concentrations from the FRM$_4$DOAS dataset. Shown here are NitroNet's monthly-mean biases averaged over all available MAX-DOAS instruments in selected altitude ranges **(a)** 0 - 200 m, **(b)** 200 - 400 m, **(c)** 400 - 600 m, **(d)** 600 - 1000 m, **(e)** 1000 - 2000 m.

e.g. in the form of the assumed profile shape by the choice of parametrization. In particular, the exponential tail of the MAPA profiles towards higher altitudes, which is the dominant characteristic here, is prescribed.

Second, computing the relative biases of NitroNet involves division of the absolute errors by the $NO_2$ concentrations of MMF, and MAPA, respectively. In the case of MAPA, these can be considerably small (e.g. $\sim 0.1 \cdot 10^{10}$ molec. $cm^{-3}$ for 1000 - 2000 m, see Fig. 9 for reference), for the reasons discussed above. Thereby, even moderate absolute errors (see right-side panel of Fig. 14) can result in large relative biases. Thus, the assessment of model performance by means of the prediction biases is informative in the lowest 3 evaluation layers (up to 600 m), but not beyond.

Another important finding of Fig. 14 is that the seasonal trends observed in Fig. 13 are represented in the lowest layer (0 - 200 m), but not the higher ones. This indicates, that the seasonal biases of NitroNet (and the underlying WRF-Chem training data) might be rooted in the lower regions of the troposphere.

## 5   Conclusions, discussion, and outlook

In this article we have introduced NitroNet, a new deep-learning $NO_2$ profile retrieval prototype for the TROPOMI satellite
instrument. NitroNet is trained on one month of RCT simulation data from the WRF-Chem model in central Europe, May 2019. The use of synthetic data allows to overcome several obstacles associated with the empirical datasets used in other studies. The main benefits of our approach can be summarized as follows:

1. Because measurements of $NO_2$ profiles are still sparse, empirical training data are effectively restricted to surface in-situ observations. A synthetic training dataset allows the neural network to learn the prediction of full $NO_2$ profiles instead.
These training profiles also cover the spatial domain continuously, and might cover scenarios which escape the in-situ observations altogether due to the strategic placement of the instruments.

2. The $NO_2$ in-situ measurements used in empirical training sets contain a hidden $NO_z$ bias of typically > 20 % due to cross sensitivities to atmospheric oxidants. Without access to model data, this bias cannot be corrected, and is silently reproduced by other neural networks.

3. The abundance of training data from the RCT simulation allows for generous dismissal of untrustworthy training examples without running into data shortage. We can therefore train the neural network on filtered data, which was purged from erroneous example profiles. The neural network can then exceed the prediction quality of the original RCT simulation.

The latter concept of "learning from the good examples, but dismissing the errors" of a data generating model was explored
in other publications (e.g. Sayeed et al. (2023); Li et al. (2023)), although in a somewhat different context. These publications describe the development of synergistic neural network + RCT combination models, while NitroNet is designed for standalone use as a surrogate model for the computationally expensive and slow RCT simulations. To put this into perspective: Using 800 CPUs, it took $\sim 5$ days to produce one month of WRF-Chem simulation data, while NitroNet can process the same amount of data in just $\sim 20$ minutes using 31 GPUs, with obvious operational advantages. Nevertheless, this functionality is limited to

the prediction of $NO_2$ profiles, and NitroNet cannot be considered a full replacement for RCT simulations, which can predict the concentrations of many other trace gases and aerosols, as well as meteorological variables.

Our main results were reported in sect. 4.2 in the form of an extensive evaluation of the NitroNet model. Three observational datasets ($NO_2$ VCDs from TROPOMI, background in-situ observations from AirBase, $NO_2$ profiles from $FRM_4DOAS$) were used as monthly-mean reference data. First, an inter-comparison between NitroNet, WRF-Chem, TROPOMI and AirBase was performed for May 2019. Hereby, the benefits of training the neural network on filtered data were demonstrated. NitroNet showed far better agreement to TROPOMI $NO_2$ VCDs than did WRF-Chem, while the comparison to AirBase surface observations returned similar results for both models. The $NO_z$ cross sensitivities of the in-situ measurements were estimated based on modelled PAN and $HNO_3$ mixing ratios, resulting in significant bias correction factors of up to +200 %.

Next, NitroNet was evaluated on previously unseen data of May 2022. The comparison to TROPOMI $NO_2$ VCDs showed a strong correlation of $R = 0.95$, a bias of $+6.7$ % and an RMSE of $2.8 \cdot 10^{14}$ molec. $cm^{-2}$. The comparison to $FRM_4DOAS$ $NO_2$ profiles showed good agreement when using the MMF retrieval algorithm (RMSE $\approx 4 \cdot 10^9$ molec. $cm^{-3}$), and slightly worse results when using the MAPA retrieval (RMSE $\approx 6 \cdot 10^9$ molec. $cm^{-3}$). The comparison to AirBase surface observations resulted in a correlation of $R = 0.75$, a bias of $-10.5$ % and an RMSE of $1.7 \ \mu g \ m^{-3}$. By omitting the instruments categorized as "urban background", the bias and RMSE were reduced to $+2.2\%$, and $1.2 \ \mu g \ m^{-3}$.

Lastly, the model evaluation was extended to different seasons (central European domain, August 2021 - July 2022) and regions of the earth (May 2022, UK, Spain and Portugal, US west coast, India, and China). Over the UK, Spain and Portugal, and the US west coast, NitroNet performed similarly well as in the original central European training domain. Over India and China, larger deviations and weaker correlations were found. The strongest differences occurred in the heavily industrialized regions of northern China, where the emission data used as model input might have been outdated. In all domains (except for the UK), NitroNet consistently overestimated the $NO_2$ load over waterbodies by approximately $10^{15}$ molec. $cm^{-2}$. The seasonal analysis revealed stable model performance in spring, summer, and early fall (March - September), but significant low biases of up to -50 % in surface concentrations during late fall and winter (October - February). Part of these underestimations may be attributed to the higher uncertainties of the main model input, the $NO_2$ VCD, during wintertime.

In closing this article, we give an outlook on future improvements and use cases of NitroNet. We will attempt to produce a full year of synthetic training data, possibly in more diverse geographical regions. This will result in more consistent model accuracy across different seasons and regions of the world. In particular, it might also help to resolve the prediction errors over water, which could be useful in addressing some of the outstanding research questions related to $NO_2$ over the oceans (e.g. the contribution of ship emissions and lightning to the lower / upper troposphere). Similarly, NitroNet could benefit from training data of higher horizontal resolution, which might improve its ability to reproduce more complex $NO_2$ profile shapes, e.g. with elevated layers. Until then, NitroNet should be considered a prototype. Furthermore, the inclusion of more data from new instruments will strongly influence the training and validation of future model versions. Here the most promising outlook is the advent of geostationary satellites, such as GEMS (see Kim et al. (2020)), TEMPO (see Naeger et al. (2021)), and Sentinel-4 (see Stark et al. (2013)). These will provide hourly resolved $NO_2$ columns, allowing for the implementation of diurnal cycles into our model. The use of more intricate MAX-DOAS retrieval algorithms could allow for better sensitivity to higher

layers of the troposphere (see e.g. Schofield et al. (2004), who achieve sensitivity to the stratosphere and upper troposphere with a zenith-sky viewing geometry). $NO_2$ profile observations from cloud-slicing (see e.g. Marais et al. (2021)) or aircraft measurements (see e.g. Riess et al. (2023); Brenninkmeijer et al. (2007)) may be used for further validation of NitroNet at various altitudes. The ongoing efforts in harmonizing observational datasets (see e.g. the GHOST dataset, see Bowdalo et al. (2024)) will allow for easier model validation at the surface in all regions of the Earth. In particular, they might open up

new possibilities to include the valuable information from surface in situ measurements into NitroNet. Previous studies have reported on neural networks trained directly on in situ observations (see e.g. Gardner and Dorling (1999); Kang et al. (2021); Chan et al. (2021); Ghahremanloo et al. (2021); Zhang et al. (2022); Jesemann et al. (2022); Cao (2023)). NitroNet aims to overcome the aforementioned disadvantages associated with empirical training targets by using synthetic training data instead. Nonetheless, information from in situ measurements could be included implicitly by using it as an additional criterion in

the data filtering procedure. This results in significantly smaller training sets, because the European in situ observations are sparse compared to the satellite measurements. Such limitations could be overcome by extension of the regional model's spatio-temporal domain, or neural network training methods specifically designed for sparse training data (e.g. by data augmentation). Lastly, more complex neural network designs, such as the invertible neural networks (INNs, see Ardizzone et al. (2018)), or physically informed neural networks (PINNs, see Raissi et al. (2019)), may be implemented once the remaining parts of the

project are deemed mature enough. This is motivated by the recent advancements in machine learning based weather forecasting (e.g. the Aurora model, based on vision transformers and encoder-decoder mechanisms, see Bodnar et al. (2024)). The NitroNet model can be used for scientific research, such as:

1. A revision of existing studies on near-surface air pollution and the associated effect on human health, with explicit treatment of the $NO_z$ biases of in-situ measurements.

2. Reprocessing of the TROPOMI $NO_2$ columns by replacing the poorly resolved $NO_2$ a priori profiles from the TM5 model (horizontal resolution: $1° \times 1°$) with the much better resolved $NO_2$ profiles from NitroNet (horizontal resolution: $3.5\,\mathrm{km} \times 5.5\,\mathrm{km}$).

3. Possibly the prediction of other trace gas profiles, such as $SO_2$ or HCHO.

Altogether, the combined efforts of machine learning, RCT modelling, and instrumental development hold promising potential

for the near future.

*Data availability.* All data is available from the authors upon reasonable request.

*Video supplement.*

*Author contributions.* LK developed the question of research under the supervision of TW and SB. LK, SO, and AP produced the training data of the neural network. LK wrote the text and produced the remaining content of the article, with all authors contributing by revising it interactively.

*Competing interests.* Some authors are members of the editorial board of the Atmospheric Measurement Techniques journal. The authors declare that they have no other conflict of interest.

*Disclaimer.*

*Acknowledgements.* We acknowledge Vinod Kumar for his invaluable help on RCT modelling, which has preceded this article. We thank Andreas Richter, Udo Frieß, Ankie Piters, Michel van Roozendael, Alexis Merlaud, Elisa Castelli, and Paolo Pettinari for maintaining the MAX-DOAS instruments and sharing their data within the FRM$_4$DOAS network. The FRM$_4$DOAS project is funded by the European Space Agency (ESA) under the contract n°4000118181/16/I-EF. Data analysis and visualization were performed using the Python programming language, including the libraries NumPy, SciPy, pandas, Xarray, matplotlib, and cartopy. The neural network of NitroNet was implemented using the PyTorch package.

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

**Table A1.** Overview of NitroNet's hyperparameters

| Hyperparameter name | Sampling range | Optimal value |
|---|---|---|
| Hidden layers | 4 - 8 | 8 |
| Neurons per layer | 100 - 400 | 326 |
| Activation function | ReLU, PReLU, CELU, GELU, SELU | PReLU |
| Loss function | MSE, $L_1$, smooth $L_1$[1], RMSLE | $L_1$ |
| Batch size | $2^9$ - $2^{13}$ | $2^{11}$ |
| Optimizer | NAdam, AdamW[2] | NAdam |
| Learning rate | $10^{-6}$ - $10^{-3}$ | $3.4 \cdot 10^{-4}$ |
| Batch normalization | True, False | False |
| Dropout probability[3] | 0 - 0.15 | 0 |
| $\Delta_{\mathrm{VCD}}$[4] | 0 - 1 | 0.2 |
| $\Delta_{\mathrm{PBLH}}$[4] | 0 - 1 | 0.1 |

For a combined reference of these terms, see Schmidhuber (2015) and Paszke et al. (2019).

[1] see the PyTorch documentation: https://pytorch.org/docs/stable/generated/torch.nn.SmoothL1Loss.html

[2] see the PyTorch documentation: https://pytorch.org/docs/stable/generated/torch.optim.AdamW.html

[3] Original range was 0 - 0.5, but training diverged for runs with dropout probability > 0.15.

[4] see sect. 3.3

## Appendix A:  Hyperparameter study

The hyperparameter study for NitroNet is based on 300 different model versions. The model configurations were sampled randomly ("random search", see Bergstra and Bengio (2012)). An overview of the hyperparameters and their respective sampling range can be found in table A1. Stochastic Gradient Descent (SGD) was not used, because all training runs using SGD diverged. The Adam optimizer was found to be out-classed by NAdam and AdamW early-on and subsequently omitted from the study. Figure A1 shows the results of the hyperparameter study in a parallel coordinate view. The validation MAPE, which is used as a performance metric to compare the model configurations, ranges from $\sim 10$ % - 30 %. This demonstrates, that a hyperparameter search can potentially improve the neural network's performance by up to a factor of 3, making it an essential step in the development of NitroNet.

## Appendix B:  Feature relevance analysis

In order to gain more insight into how the neural network of NitroNet operates, a feature relevance analysis was conducted. The goal is to quantify, how strongly each input variable contributes to the overall model performance. The standard method is to compute the *Shapley scores* of the input variables (see Shapley (1951)). The Shapley score of the $i$-th input variable $x_i$ is

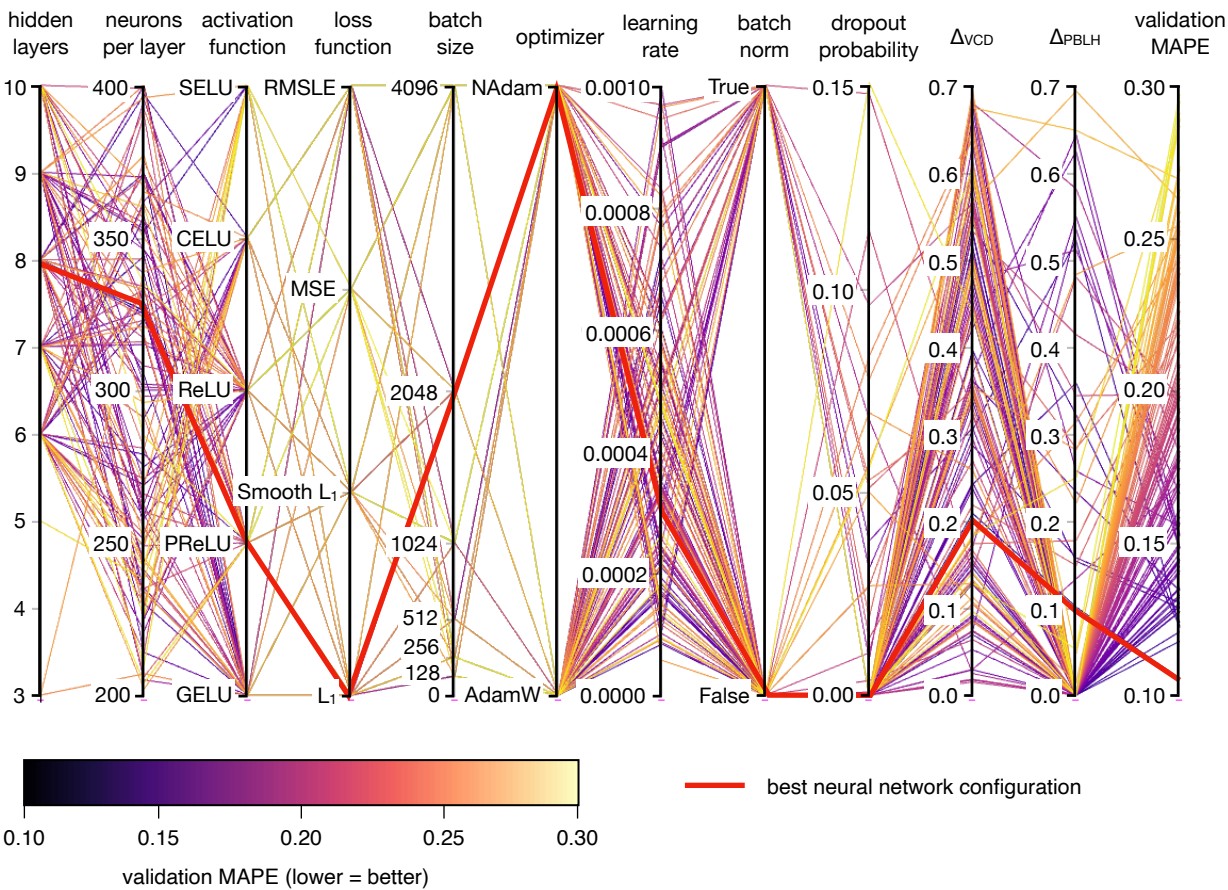

**Figure A1.** Results of the hyperparameter study in a parallel coordinate view. Each hyperparameter is represented by one vertical axis ("hidden layers", "neurons per layer", ...). Each variant of the neural network is represented by a contiguous line, intersecting the vertical axes at the network's hyperparameter values. The last vertical axis shows the MAPE achieved on the validation set, which acts as the metric for the selection of the best neural network configuration (lower = better). The optimal configuration is drawn as a thick red line.

defined as

$$R_i = \sum_{S \subseteq P \setminus \{x_i\}} \frac{|S|!(|P| - |S| - 1)!}{|P|!} \left( f\left(S \cup \{x_i\}, y_{\text{true}}, y_{\text{pred}}\right) - f\left(S, y_{\text{true}}, y_{\text{pred}}\right) \right) \tag{B1}$$

where $P$ denotes the set of all input variable variables, and $|\cdot|$ the set cardinality. $f(I, y_{\text{true}}, y_{\text{pred}})$ is a function of choice, which acts as a measure for model performance by comparison of the ground truth $y_{\text{true}}$ vs. the model's predictions $y_{\text{pred}}$, using either all input variables (i.e. $I = S \cup \{x_i\}$) versus omitting variable $x_i$ (i.e. $I = S$). Omission of an input variable $x_i$ is simulated by replacing its values with with random samples from the validation set (approximating a sample drawn from the prior probability distribution of $x_i$.). The "feature relevance" $F_i$ is obtained by normalization of the Shapley scores, i.e. $F_i = R_i / \sum_i R_i$. The following further premises were made:

1. We define

$$f = \frac{\text{RMSE}(I, y_{\text{true}}, y_{\text{pred}}) - \text{RMSE}(S = \emptyset, y_{\text{true}}, y_{\text{pred}})}{\text{RMSE}(S = P, y_{\text{true}}, y_{\text{pred}}) - \text{RMSE}(S = \emptyset, y_{\text{true}}, y_{\text{pred}})} \tag{B2}$$

   i.e. we we use a scaled RMSE to measure model performance. The "uninformed" case (omitting all input variables, $I = \emptyset$) equates to a model performance of $f = 0$, and the "fully informed" case (omitting none of the input variables, $I = P$) equates to a model performance of $f = 1$. Subsequently, all Shapley scores lay in the interval $[0, 1]$.

2. Because the sum in eq. (B1) iterates over a power set of large cardinality, not all summands can be evaluated. Instead, $R_i$ is approximated by computing random summands of eq. (B1) until the overall distribution of the feature relevances has converged.

3. Certain input variables are grouped together (e.g. the group "wind" contains all wind speed variables and does not discriminate between $u$ and $v$ direction).

The feature relevance can also be computed separately for each vertical layer. The resulting feature relevance profiles are shown in Fig. B1. We draw the following conclusions:

1. The $NO_2$ VCD is generally the most important input variable from 0 to 1500 m altitude.

2. The feature relevance of the PBLH peaks at $\sim 1800$ m, which corresponds to the average PBLH value in WRF-2019. Because the $NO_2$ profiles show strong gradients at the top of the PBLH, this feature relevance profile shape is expected.

3. The $NO_2$ concentrations above the PBL are known to be low and weakly correlated to satellite observations. Here, the model performance is dominated by the input groups "surface class" and "tropospheric AMF", which the neural network most likely uses to predict average $NO_2$ profile estimates, based on coarse general constraints (e.g. "over water", "rural land", "urban land").

4. At the surface, there is a trade-off between the feature relevance of emission data and the $NO_2$ VCD. This confirms that emission data are a valuable addition to NitroNet, as they can improve the model performance by almost 20 %.

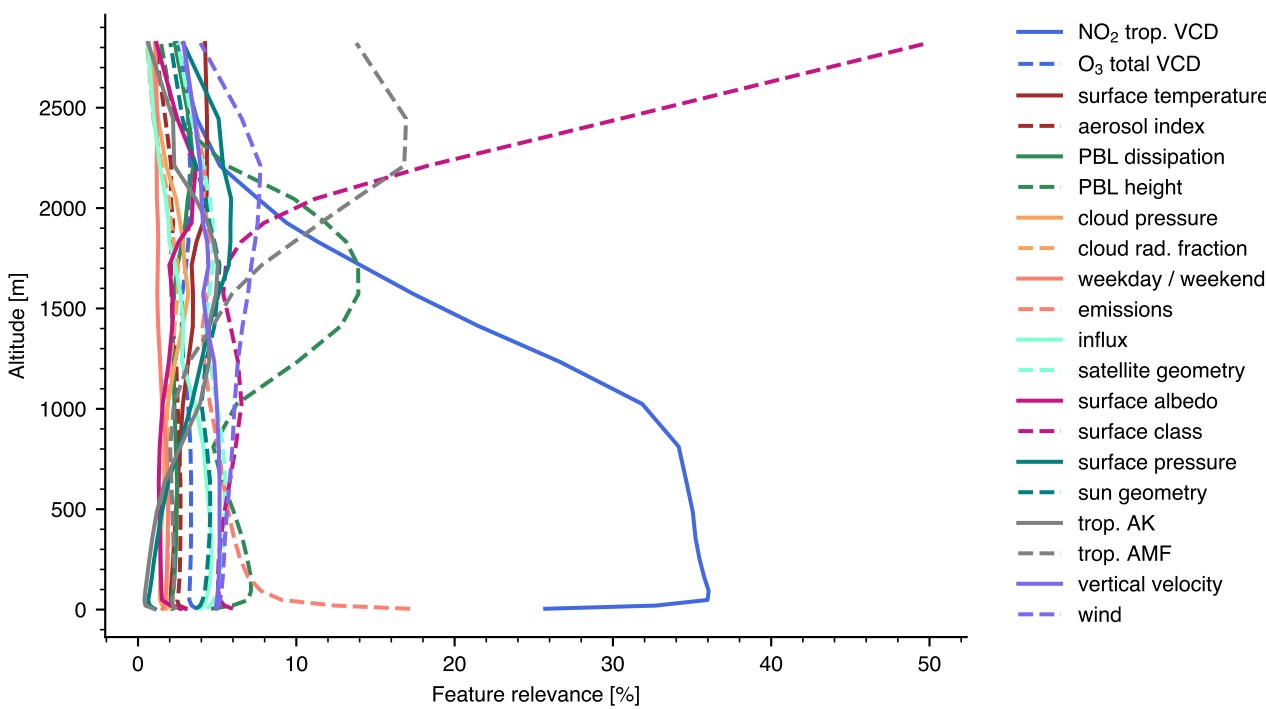

**Figure B1.** Vertically resolved feature relevance analysis of the NitroNet model.

The feature relevance of the emission data is further demonstrated in Fig. B2. Comparing Fig. B2a and B2b shows, that when no emission data is used, NitroNet's prediction of the $NO_2$ surface concentration is essentially proportional to the $NO_2$ VCD. Once emissions are added as input (see Fig. B2c), the distribution of predicted surface concentrations becomes significantly more complex: High values suddenly occur despite of comparably low VCDs (e.g. in the cities of Hamburg and Berlin, Germany) and fine-scale infrastructure, such as car highways connecting cities, becomes visible.

**Appendix C: Additional figures**

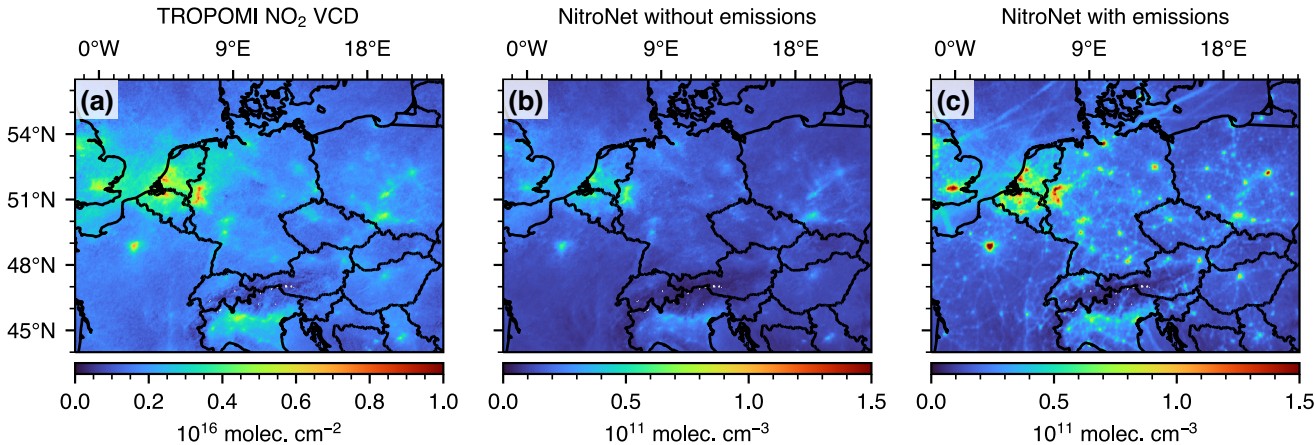

**Figure B2.** Demonstration of the "emissions" feature relevance. Subplot **(a)** shows the monthly-mean $NO_2$ VCD from TROPOMI (May, 2019). Subplots **(b)** and **(c)** show the corresponding $NO_2$ surface concentration from NitroNet with all emissions turned off / on, respectively.

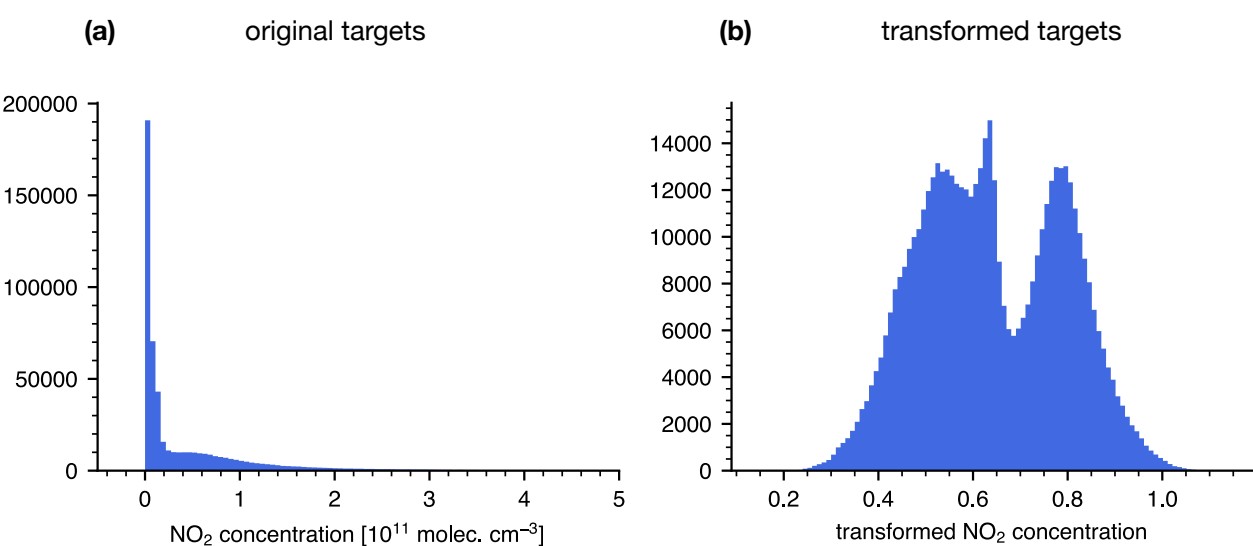

**Figure C1.** Example of the data transformations used during the training of NitroNet. Shown here: histograms of the training targets ($NO_2$ concentrations) at all altitudes before **(a)** and after **(b)** application of a logarithmic data transformation. The transformed targets are unitless.

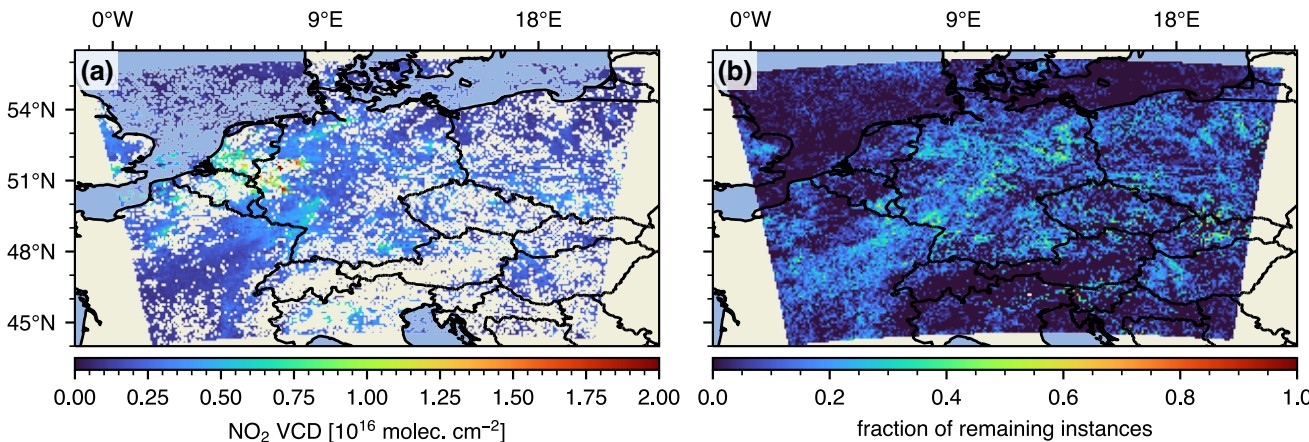

**Figure C2.** Overview of the TROPOMI $NO_2$ VCDs (with re-computed air mass factors) upon application of the data filter described in sect. 3.3. **(a)** shows the remaining data, averaged over all orbits of May 2019. **(b)** shows the remaining fraction of instances in relation to the un-filtered dataset.

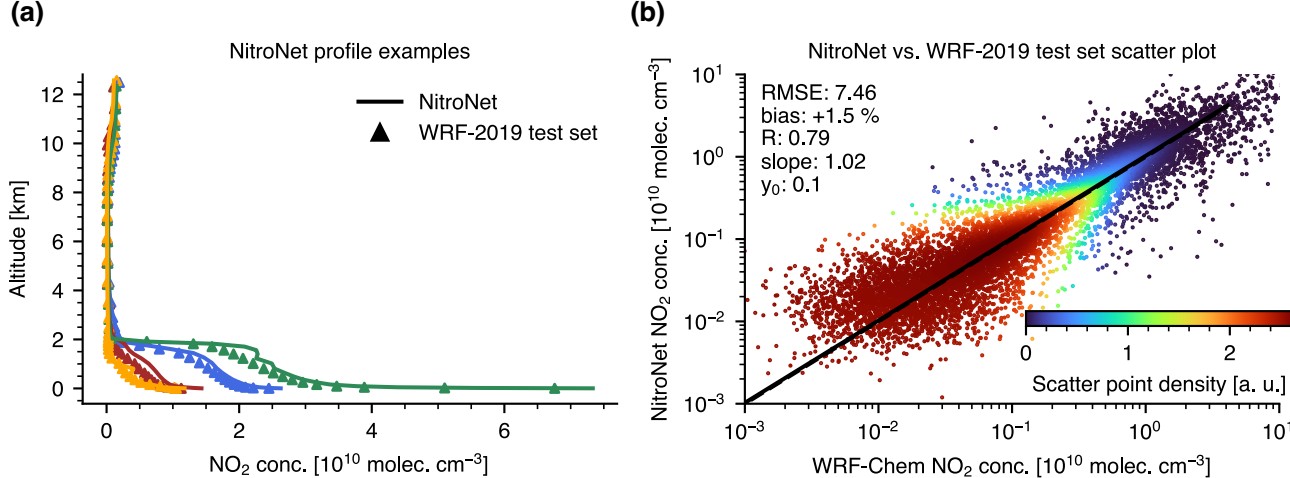

**Figure C3.** Like Fig. 3, but computed on the un-filtered test set.

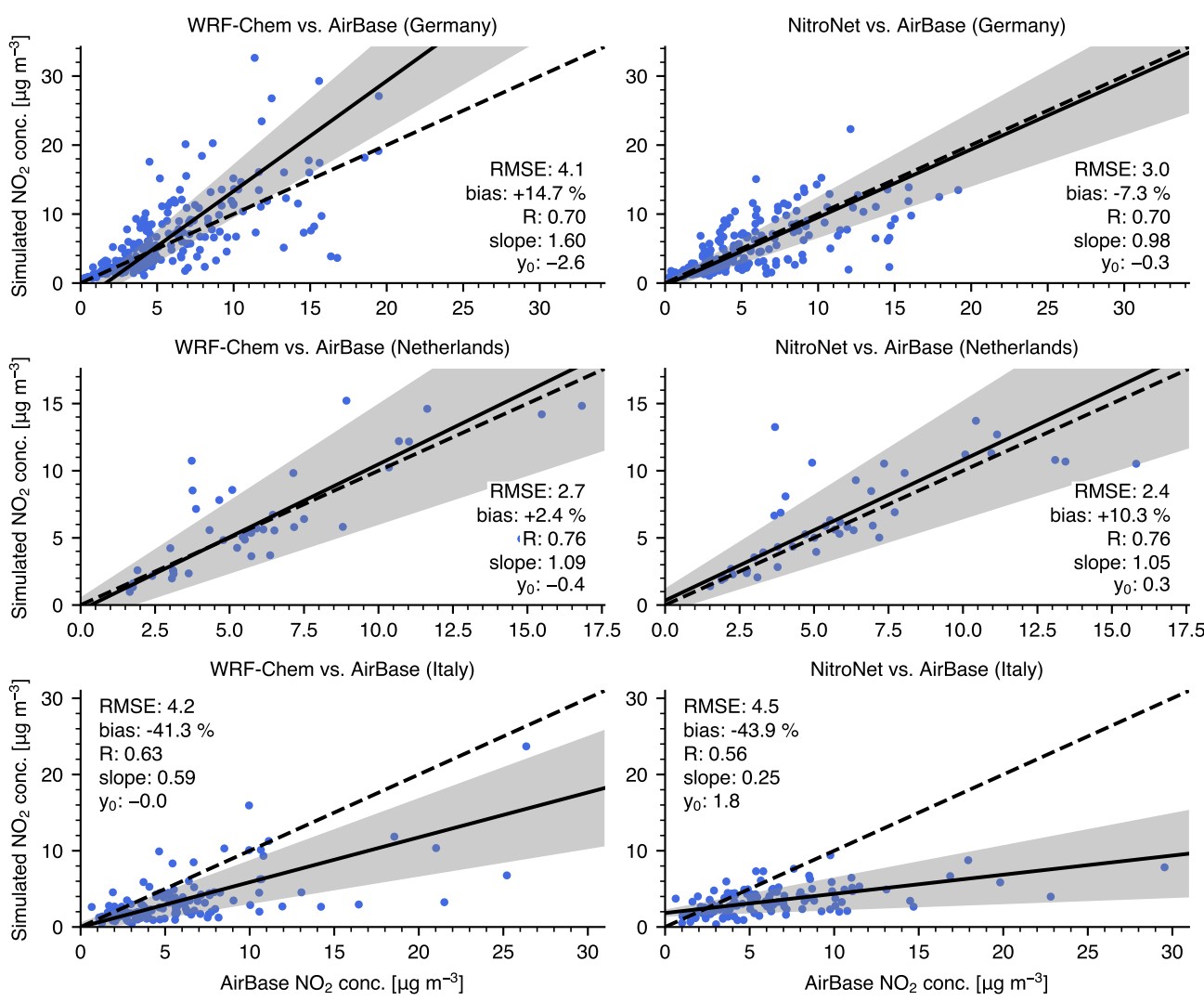

**Figure C4.** Scatter plots of the data shown in Fig. 5 restricted to individual countries (Germany, Netherlands, and Italy).

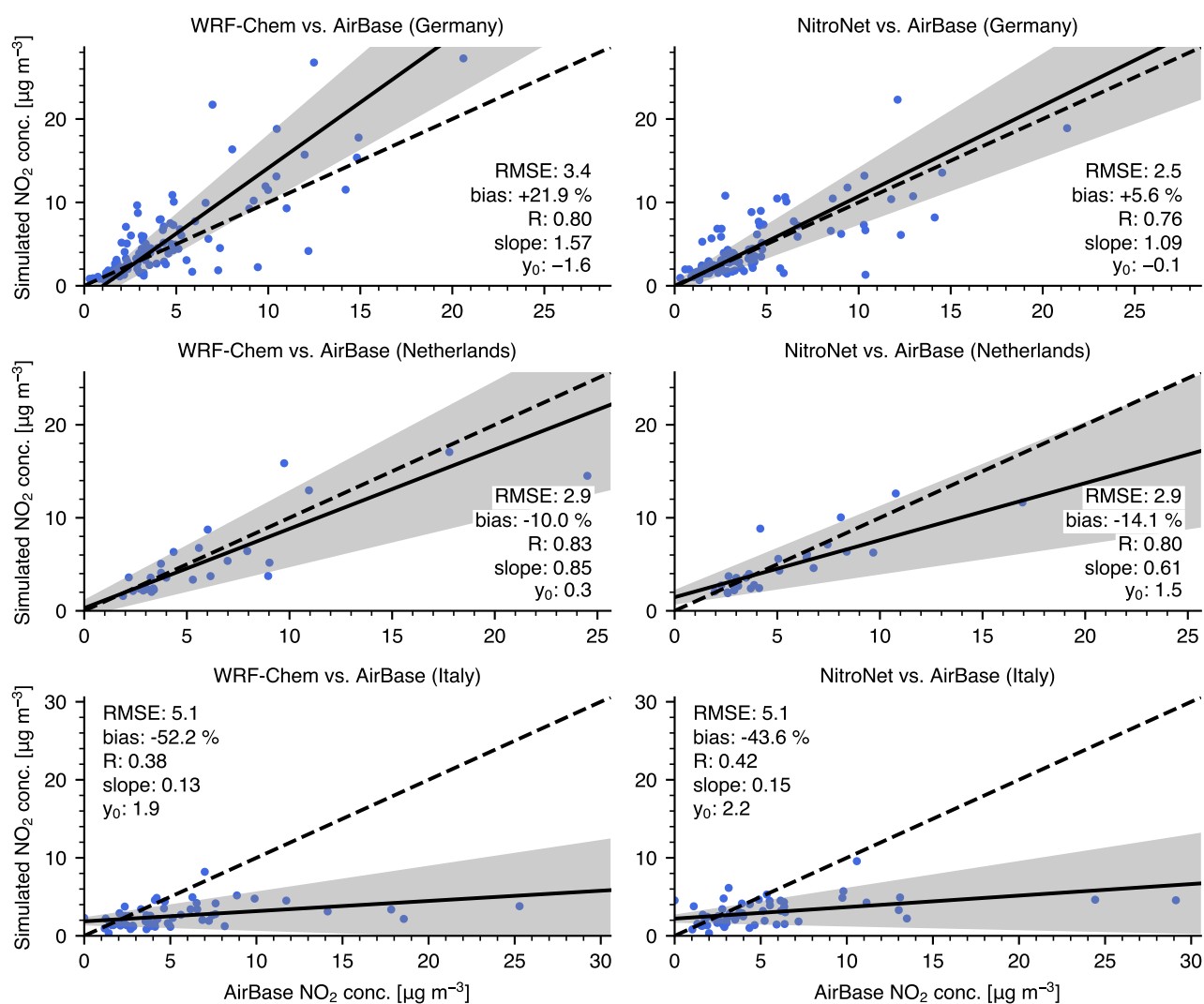

**Figure C5.** Like Fig. C4, but without urban stations.

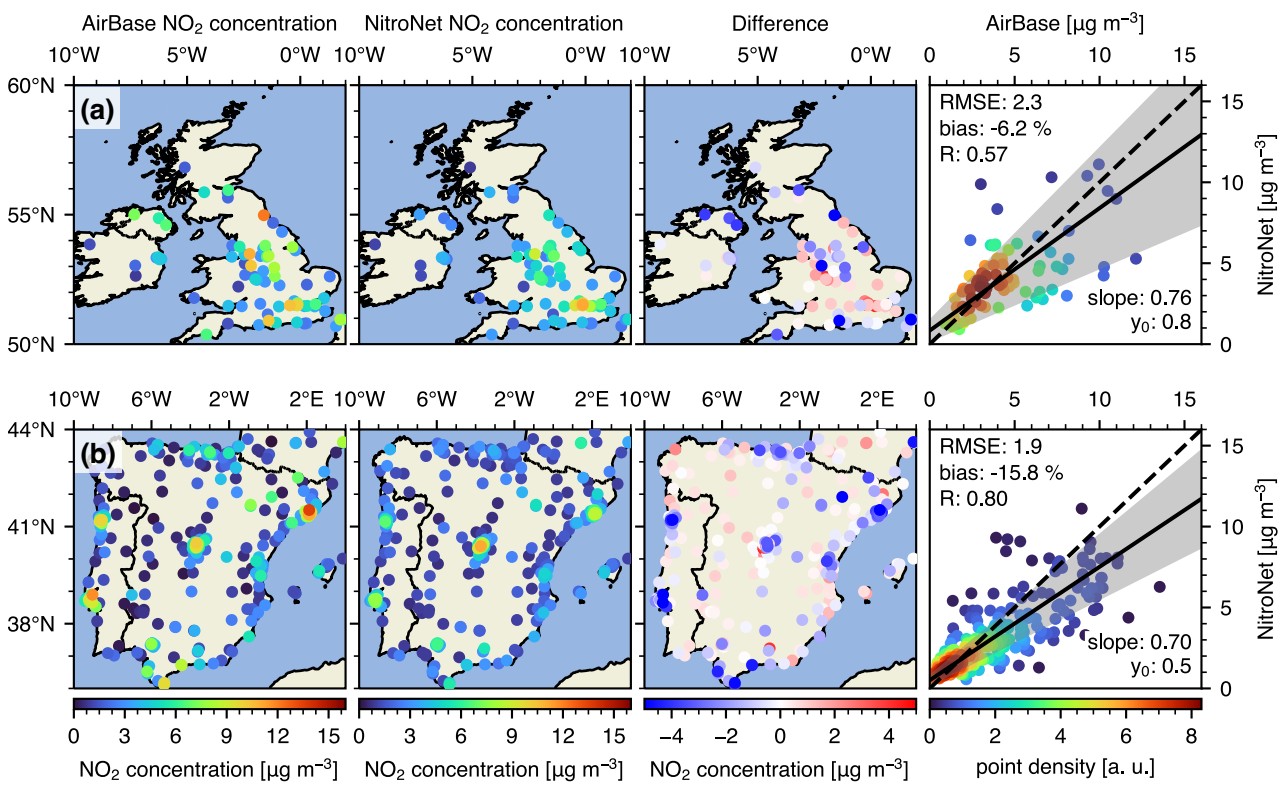

**Figure C6.** Like Fig. 11, but with urban stations included. RMSE and intercept are displayed in $\mu g\ m^{-3}$.

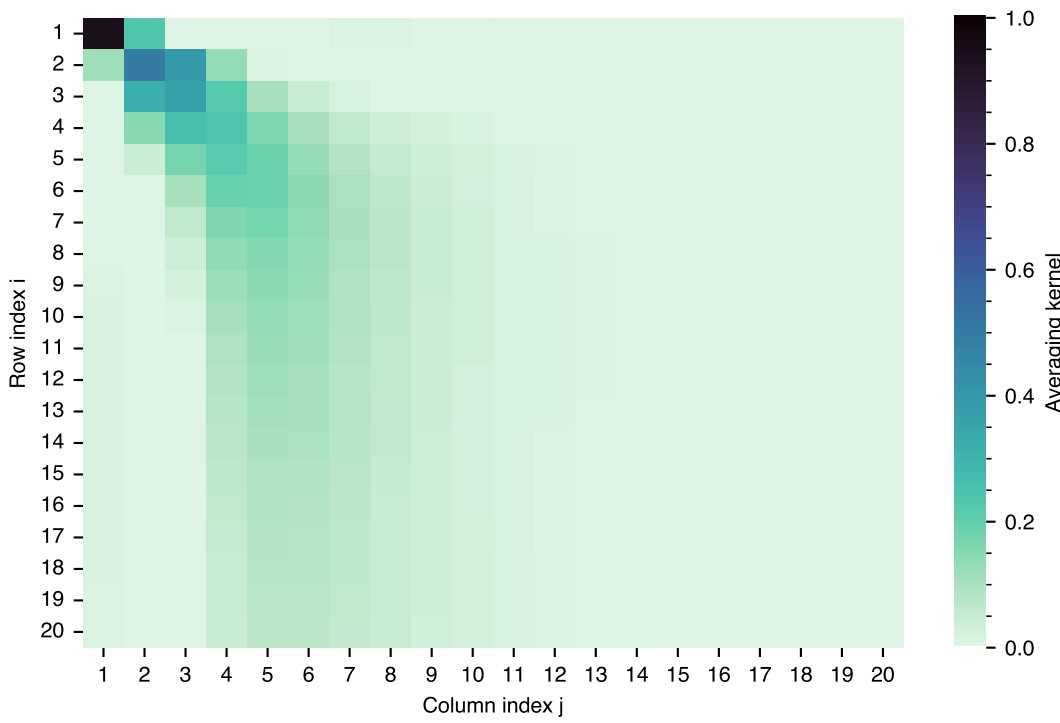

**Figure C7.** Monthly-mean averaging kernel matrix from the FRM$_4$DOAS instrument in Heidelberg, May 2022. The rows and columns are ordered such that index 1 represents the lowest layer of the retrieval, and index 20 the highest. Each layer has a vertical extent of 200 m.

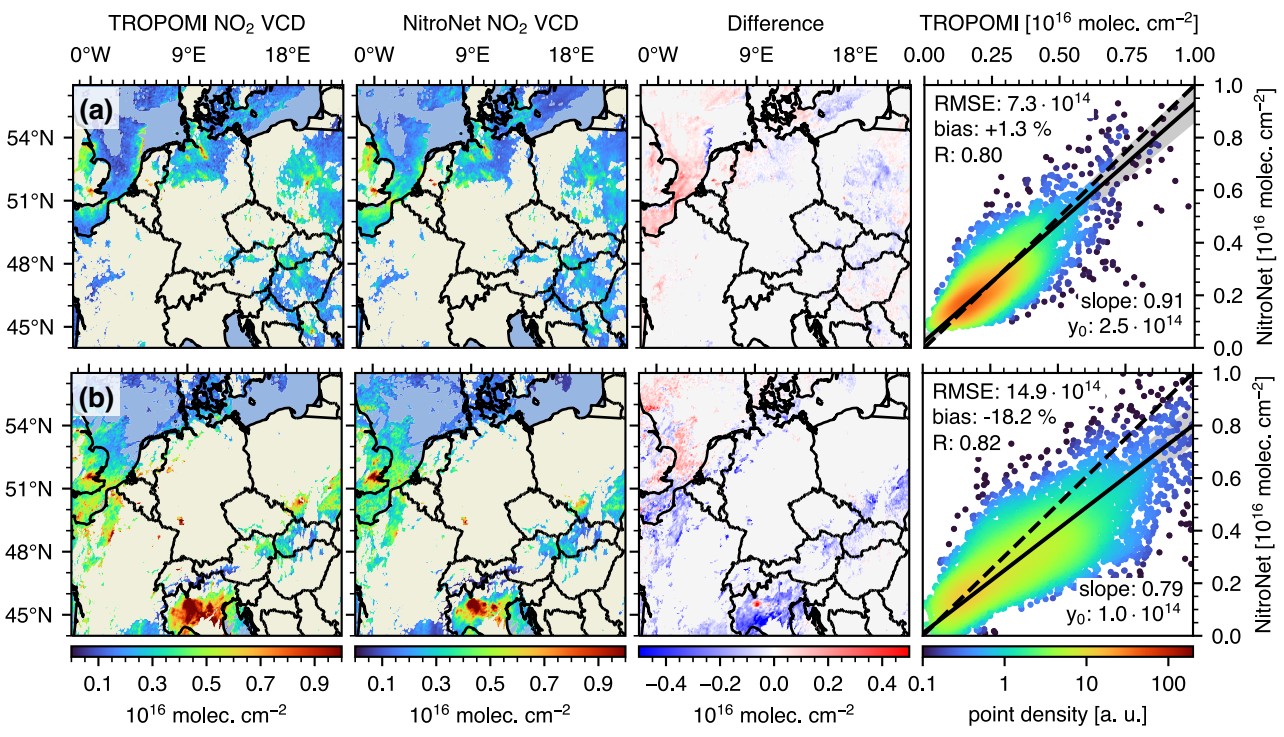

**Figure C8.** Like Fig. 7a, but for two single summer and winter days. **(a)** shows data of 2022-05-05. **(b)** shows data of 2021-11-05.

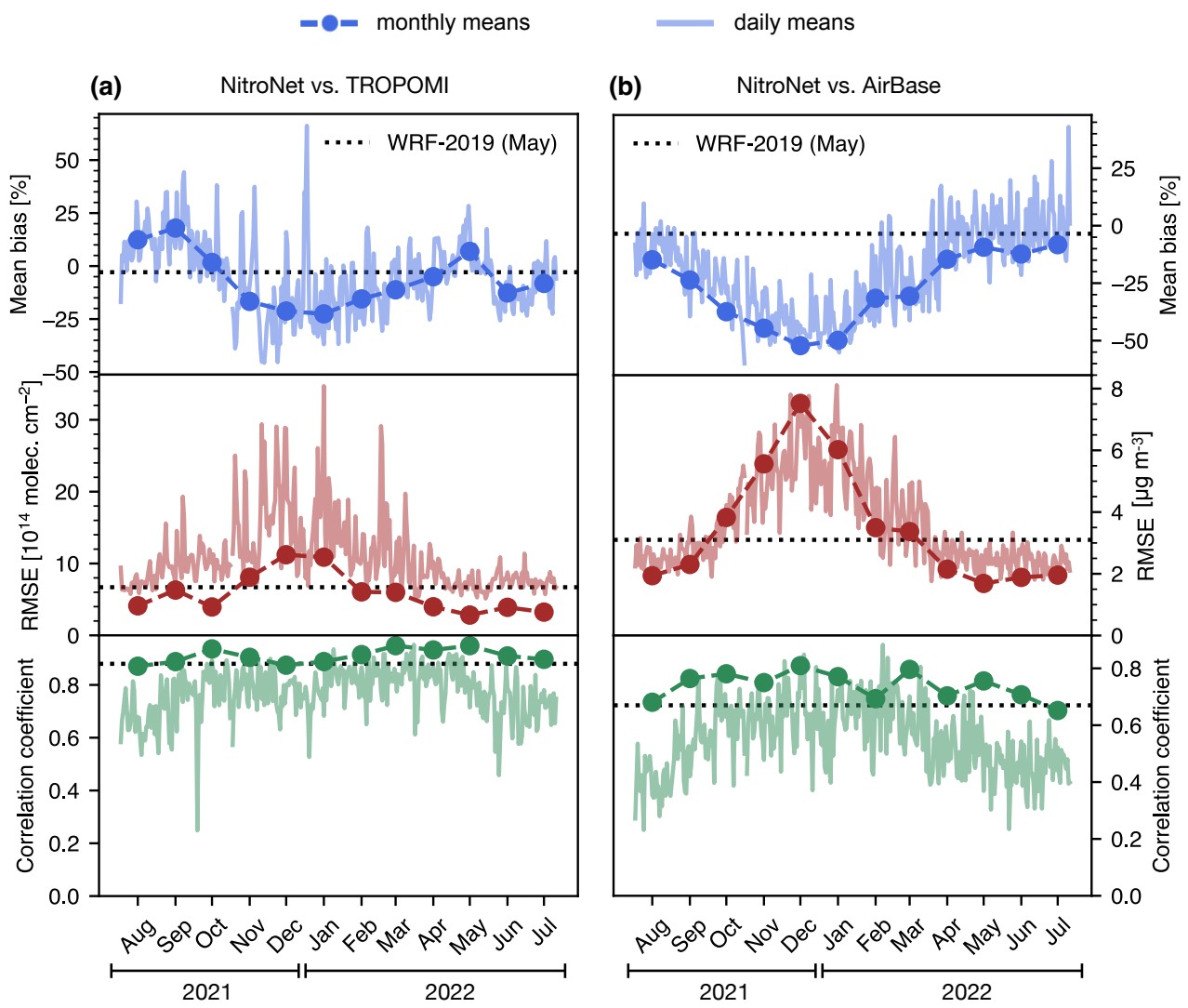

**Figure C9.** Like Fig. 13, but with urban stations included.