# Peer review of "NitroNet - A deep-learning $NO_2$ profile retrieval prototype for the TROPOMI satellite instrument"

_EGUsphere, 2024_

## Author Response (AR1)

**Authors' reply to RC1 by Anonymous Referee 1**

We would like to thank Anonymous Referee 1 for the constructive and encouraging review of our manuscript. Please find our point-by-point replies below. The Referee's comments are printed in blue, and our replies in black.

General comments:

1. Line 102: Using only one month as training data cannot enable the model to learn seasonal variability, and the authors also acknowledge this limitation in lines 389-390. It is suggested to clarify the reason for choosing only one month in the input data preparation sections, which would be useful for the readers.

We acknowledge, that this point should have been made more clear.

Ideally, we would like to train on a full year of data. However, the process of generating the training data is not straight forward. Models like WRF-Chem often show severe disagreement to reference data (e.g. $NO_2$ VCDs from TROPOMI or the AirBase in situ measurements). This was also the case for our training data of May 2019. However, we were able to fine-tune some of the model's parameters, which has lead to a significant reduction of the discrepancies between the model and observational data. This process took approximately one year, and is documented in detail in our previous publication, see Kuhn et al. (2024). Unfortunately, simulation setups that work well for summertime do not necessarily work well for wintertime. In test runs of our simulation for winter months, we found large model biases in the form of $NO_2$ VCD overestimations by + 50 %. This is similar to the findings of Douros et al. (2023) and others.

Altogether, our WRF-Chem simulation setup can currently not provide what the Referee asks for. This might be resolved in the future, but is not within the scope of our article and should be expected to take a long time.

The following paragraph was added to sect. 2.1: *This study revolved around the question, how certain WRF-Chem model parameters can be optimized in order to improve the model's agreement to various reference datasets. In particular, optimization of the model's vertical mixing parametrization was found to be crucial to improve the agreement to in situ observations of surface $NO_2$. Unfortunately, such optimization processes take a long time to solve if the underlying model is as computationally expensive as WRF-Chem. Additionally, wintertime RCT simulations are known to be particularly challenging (see e.g. Douros et al. (2023)), mainly due to their tendency to overestimate the total $NO_2$ columns severely. Therefore, full-year training data with a resolution and accuracy compared to our summertime data cannot be provided for now. Although NitroNet was trained exclusively on summertime data, it can be used in other seasons as well, although with larger prediction errors (as discussed in sect. 4.3).*

2. Section 2.1: As the NO2 is mainly present at the near surface and much less in the upper layers of the atmosphere, the large difference in the magnitude of the NO2 between the layers can lead to a skewed distribution of the targets in the training data set. In this work, a feature transformation was applied to the input variables, but it is not clear whether the transformation was also applied to the target variable. If not, would this affect the predictive ability of the model in the higher layers? It would be beneficial to clarify this. It is also suggested to clarify the data splitting strategy (e.g. sample-based, space-based, or period-based).

The Referee's observation is correct. The original target distribution is skewed. Therefore, a logarithmic transformation was applied to the targets, resulting in a less skewed distribution (see Figure C1 below). This is expected to improve the model's performance overall, independent of the altitude.

Figure C1 was added to the Appendix of the manuscript.

[Figure]

**Figure C1.** Example of the data transformations used during the training of NitroNet. Shown here: histograms of the training targets ($NO_2$ concentrations) at all altitudes before **(a)** and after **(b)** application of a logarithmic data transformation. The transformed targets are unitless.

The following paragraph was added to sect. 3.2: *Feature transformations are known to improve the predictive capability of machine learning models, particularly if the features or targets have a skewed or long-tailed distribution. This is the case for some of NitroNet's input features (e.g. the $NO_2$ VCD). Likewise, transformations are applied to NitroNet's training targets (the $NO_2$ concentrations at different altitudes), see e.g. Fig. C1.*

A sample-based splitting strategy was used. This means, that the training, validation, and test data are obtained by drawing a corresponding number of samples from the full dataset without replacement.

The following sentence was added to sect 2.1: *The partitioning is obtained by unweighted random sampling without replacement.*

3. It is suggested that section 3.1 be merged with section 2.2 as both sections describe the model input.

We prefer to keep the structure of section 3.1 and section 2.2 as is, for the following reason:

Section 2 describes the properties of the datasets used in our article. These are independent of our model design, choices in training procedure, and methodology.

Section 3.1, on the other hand, is highly specific to our design choices. Examples are the classification of SNAP sectors, the choice of ERA5 model levels, the ternary surface classification, etc.

We find that this is an important difference that justifies separate sections, particularly if they follow one another directly.

The following sentence was added to sect. 3.1: *In contrast to sect. 2, the descriptions given here are based on our design choices, e.g. how the used data were selected and processed.*

4. Section 3.3: It is necessary to explain more about the filtering strategy. The filtering takes the TROPOMI data and the ERA5 PBLH data as reference data, but their uncertainty should also be acknowledged here. Meanwhile, it would be beneficial to show the spatial distribution of the filtered

*training samples to check if there are still enough samples left for different grids within the study area. For example, the number or proportion of training samples left for a grid.*

The text was changed to acknowledge the uncertainty of the reference data. The following paragraph was added to sect. 3.3: *Figure C2 gives an overview of the spatial distribution of NO$_2$ VCDs after filtering, and the fraction of remaining instances across the domain.*

*It should be mentioned, that the TROPOMI NO$_2$ VCD and the ERA5 PBLH are quantities with significant uncertainties. For the retrieval of the tropospheric NO$_2$ VCD, the tropospheric air mass factor uncertainty (typically 20 % - 50 %) is known to dominate the overall uncertainty of the column (typically 30 % - 60 %), see e.g. Liu et al. (2021). Guo et al. (2024) report summertime ERA5 PBLH errors of approximately 150 m over continental regions, derived from radiosonde measurements. With an average PBLH of approx. 1500 m on the WRF-2019 domain, this amounts to a relative uncertainty of approx. 10 %.*

A plot of the monthly-mean NO$_2$ VCDs in the filtered training data set can be found in panel (a) of Figure C2 below.

[Figure]

**Figure C2.** Overview of the TROPOMI NO$_2$ VCDs (with re-computed air mass factors) upon application of the data filter described in sect. 3.3. **(a)** shows the remaining data, averaged over all orbits of May 2019. **(b)** shows the remaining fraction of instances in relation to the un-filtered dataset.

Because the filter criteria are very strict, the filtering does result in gaps in the spatial distribution of the targets. Panel (b) shows the proportion of remaining training samples in each location. The Referee's comment suggests, that a reduced number of samples would have a negative impact on the neural network. This could be a valid concern, if the filtering removed entire retrieval scenarios (e.g. all water pixels), but this is not the case here. Figure C2 was added to the Appendix of the manuscript.

5.   Lines 247, 252: It's questionable whether the validation is fair here, as the test dataset is also filtered. The validation and test data sets are very important for examining the generalization performance of the model and should represent the unseen scenario to which the NitroNet is applied. The validation data is used to optimize the model and the test data is used as a final check. Considering that the model is not only used within the training month (i.e., May 2019), the filtering based on WRF results is not available when the model is applied to a different area and period. The current test result is likely to give an overly optimistic assessment of the model generalization. Therefore, it is suggested to complement the comparison based on the unfiltered test dataset.

The Referee is correct. The test data shown in Fig. 3 and discussed in l. 245 - 255 is filtered. It is also correct, that when the filter is removed, there are larger errors in the neural network prediction. Nonetheless, the comparison against a filtered test set is both more informative, as well as methodically correct for the following reasons:

1)  If the validation data were not filtered, the model's hyperparameter optimization would be influenced by the erroneous data of the WRF-Chem simulation.

2) If the test data were not filtered, it would mask a significant property of the neural network, namely its robustness towards (some) systematic errors of the WRF-Chem model. As demonstrated, using filtered training data can prevent the neural network from reproducing the severe overestimations of $NO_2$ in western Europe. If we use unfiltered test data, we would compare NitroNet predictions to erroneous WRF-Chem examples in many cases, meaning that deviations between the two would actually be favorable. This skews the interpretation of the study.

3) Using filtered test data allows to investigate the neural networks generalization capability in isolation. This way, we can assess how well the neural network reproduces unseen test data of exactly the same distribution as the training data in absence of any other effects. It tells us, what portion of the NitroNet errors revealed in the validation study later are due to the choice of neural network and its training procedure, e.g. the „model misspecification error", and the „model optimization error".

Using un-filtered test data would only serve one purpose: to (implicitly) assess the performance of NitroNet in „real" use-cases when no data filtering can be applied. However, this is already investigated explicitly, in sect. 4.2.

A version of Fig. 3 showing the evaluation on un-filtered data is shown below in Fig. C3:

[Figure]

**Figure C3.** Like Fig. 3, but computed on the un-filtered test set.

Figure C3 was added to the Appendix of the manuscript. The following paragraph was added to sect. 4.1:
*Note, that Fig. 3 shows data from the filtered test set exclusively. This choice was made for two main reasons: On one hand, we aim to exclude supposedly erroneous $NO_2$ profiles from WRF-Chem for the evaluation of NitroNet. These would result in larger errors in the comparison between WRF-Chem and NitroNet, particularly because the WRF-Chem $NO_2$ profiles show systematic errors that NitroNet does not reproduce. This is demonstrated more explicitly further below. On the other hand, the evaluation against filtered test data is an assessment of the neural network's performance in isolation, i.e. it indicates its prediction errors on instances from the same distribution as the training set. For completeness, a version of Fig. 3 based on un-filtered test data is shown in Fig. C3.*

6. Lines 335-336: Omitting the urban station from the validation does not seem to be a good choice. The dynamic of NO2 is closely related to human activity, while the measurements outside the urban stations mainly provide low and relatively stable NO2 Comparing the model results with these non-urban stations, it is difficult to evaluate the ability of NitroNet to capture the surface NO2 dynamics.

We acknowledge, that this section of our article should have been explained more clearly.

On one hand, the slight low bias of NitroNet in comparison to urban stations is an important finding that we decided to show in our article. On the other hand, we have reasons to be skeptical towards the stations' classification, as described in l. 327-336. In response to the Referee's comment we have added another argument to the corresponding discussion.

The following paragraph was added to sect. 4.2.1: *Furthermore, Fig. 7 shows significant low biases in NitroNet's surface predictions, but no corresponding low biases in the tropospheric columns. This can partly be attributed to the inter-pixel variability of the TROPOMI measurements. Surface stations with a large NitroNet bias are possibly located closer to strong traffic emissions, and thus less correlated with the $NO_2$ VCD, which acts as the main model input.*

Besides that, we believe that the best way to convey these ambiguities to the reader is to always show both versions (with / without urban stations) of each figure. The following figures are affected (figure names refer to the revised manuscript version):

- Figure 7: already shows both versions

- Figure 11: a version with urban stations is already shown in Fig. C6

- Figure 13: a version with urban stations was added (Fig. C9)

- Figure C4: already shows urban stations. A version without urban stations was added (Fig. C5).

7. Lines 425-450 and Figure 13: This part of the study examines the seasonal performance of the NitroNet on a daily and monthly basis, with a particular focus on the monthly-mean results. Figure 13 illustrates that while both results exhibit comparable trends, the monthly-mean results are superior to the daily-mean results. Furthermore, the monthly-mean results are not simply the average of the daily-mean results. Considering that the NitroNet provides hourly outputs and the TROPOMI overpasses once a day, it is more realistic to report the daily means in practice. The use of monthly means may result in overly optimistic assessments, and authors also recognize that the use of monthly means can reduce statistical noise, as stated in line 449. It can be understood that the significant data gap interferes with the daily-mean results as stated in lines 446-449. Consequently, it is recommended that the authors aggregate all the daily means and then calculate the overall RMSE, bias, and Pearson correlation as a monthly evaluation.

If the metrics are computed on monthly-mean data, they are indeed overly optimistic, because the averaging reduces the noisiness of the NitroNet predictions. We would still like to show the monthly-mean comparison in order to maintain inter-comparability to other model studies, some of which have shown monthly-mean results in the past. This requires we leave the evaluation as is. Please note, that the monthly means of the diagnostics computed on daily data can be directly obtained from Fig. 13 as shown. This is not the case for the diagnostics computed on monthly means. The explanations were made more clear in order to avoid ambiguities.

The following paragraph was added to sect. 4.3: *The NitroNet predictions were evaluated against TROPOMI and AirBase observations, and time series of the bias, RMSE, and correlation coefficient were computed, see Fig. 13. Shown here are daily mean values, as well as monthly mean values in analogy to the other evaluations presented up to this point. Note, that in this context "monthly-mean bias" refers to the bias computed on monthly means, as opposed to the monthly mean of daily biases (which can be estimated from the daily values shown in Fig. 13). The same holds for the RMSE and the correlation coefficient. Because averaging over multiple days reduces the noisiness of the NitroNet predictions, the monthly-mean RMSE values are smaller, and the correlation coefficients larger, than on unaveraged data. The mean biases, however, are unaffected by averaging.*

8. Section 2, conclusions, discussion, and outlook: Have the authors considered using surface NO2 from in-situ measurements as another target for model training? It is not necessary to emphasize the uniqueness of NitroNet over previous models by training it only on NO2 There are hundreds (even thousands) of stations measuring surface NO2 which is the dominant fraction of total NO2. Thus, using measured surface NO2 as an additional target could enrich the constraints for model training and improve model agreement with surface measurements. Although the authors point out the uncertainty inherent in the in-situ measurements several times, the uncertainty in the synthetic data and the remote sensing data cannot be ignored either. Therefore, it is better not to underestimate "training on ground measurements". The prospects of incorporating ground data into NitroNet training should be discussed.

Training neural networks on targets from surface $NO_2$ measurements has become a popular topic with many published articles over the past years. With the methodological framework laid out, the newer studies mainly vary by type of neural network, geographic region, and the input variables. However, using empirical

training targets is linked to irresolvable limitations (e.g. the lack of profile data, and the presence of $NO_z$ biases in the training data). To explore how these can be overcome by synthetic training data is among the main topics of our article. Therefore, training NitroNet on $NO_2$ surface measurements as targets is not in its scope.

In the future, we might explore methods to use the agreement between model-borne training data and surface in situ measurements as an additional data filter criterion. This would (implicitly) embed some of their information into NitroNet. However, this is currently infeasible, because it reduces the total number of remaining profiles far too much (mostly due to the sparsity of the measurements). This might change, e.g. by extension of the spatio-temporal model domain, or by using machine learning methods particularly designed for sparse training data.

The following paragraph was added to sect. 5: *The ongoing efforts in harmonizing observational datasets (see e.g. the GHOST dataset, see Bowdalo et al. (2024)) will allow for easier model validation at the surface in all regions of the Earth. In particular, they might open up new possibilities to include the valuable information from surface in situ measurements into NitroNet. Previous studies have reported on neural networks trained directly on in situ observations (see e.g. Gardner and Dorling (1999); Kang et al. (2021); Chan et al. (2021); Ghahremanloo et al. (2021); Zhang et al. (2022); Jesemann et al. (2022); Cao (2023)). NitroNet aims to overcome the aforementioned disadvantages associated with empirical training targets by using synthetic training data instead. Nonetheless, information from in situ measurements could be included implicitly by using it as an additional criterion in the data filtering procedure. This results in significantly smaller training sets, because the European in situ observations are sparse compared to the satellite measurements. Such limitations could be overcome by extension of the regional model's spatio-temporal domain, or neural network training methods specifically designed for sparse training data (e.g. by data augmentation).*

Minor comments:

1. Line 14: Here the authors mention three datasets, so the statistics from MAX-DOAS validation should also be mentioned. Changed as proposed.

2. Line 16 "summertime": Maybe the "late spring" would be more appropriate. Changed as proposed.

3. Lines 54-56: The typical uncertainty of MAX-DOAS is suggested to be mentioned.

The following sentence was added to sect. 1: *An intercomparison study of MAX-DOAS retrieval algorithms by Tirpitz et al. (2021) revealed relative retrieval uncertainties of between 3 % and 70 %, which can be expected to be the dominant part of the total MAX-DOAS uncertainty.*

4. Lines 85-92: The statement of the benefits of the NitroNet approach could be further refined. Here are two points the authors might refer to:

    1. Leveraging the synthetic data can overcome the limitations of insufficient measurements and enable the ML model to perform the prediction task for a larger space with more dimensions.

    This was already expressed in l. 85-86 (original manuscript).

    2. In addition to providing a substantial number of training samples, the synthetic dataset allows the model to learn a more general and physically plausible pattern of NO2, which could enhance the model's generalization performance.

    It should not be claimed that the 3D distributions of $NO_2$ from a WRF-Chem simulation are generally more physically plausible than actual measurements. One might argue that this is the case, if the $NO_z$ biases are taken into consideration. This, however, was already expressed in l. 87-88 (original manuscript).

5. Lines 120-122: Although QA filtering is a common operation for TROPOMI data processing, it may result in significant missing data and limit the generalization of the ML model. As shown in Figure C4, such filtering may result in few samples over NO2 hot spots for model training. Moreover, does it mean the NitroNet can only be used when TROPOMI QA is greater than 0.75? A related discussion is suggested.

NitroNet can be used with arbitrary qa-values, but a threshold of qa > 0.75 is recommended. Please refer to Fig. C2 above, which shows only data with qa > 0.75.

6.  Lines 124-125: ERA5 hourly reanalysis data resolution should be 0.25 rather than 0.125. The ERA5-land hourly has the resolution 0.1 but seems not used in this work. Meanwhile, the year of reference provided seems to be 2016 instead of 2017 (http://www.ecmwf.int/en/newsletter/147/news/era5-reanalysis-production).

Many thanks for the hint! This mistake on our behalf has been corrected, and the reference was changed to:

Hersbach H, Bell B, Berrisford P, et al. The ERA5 global reanalysis. Quarterly Journal of the Royal Meteorological Society, 146, 1999–2049, https://doi.org/https://doi.org/10.1002/qj.3803, 2020.

7.  Lines 144 and 194: As this work mentions the Monte Carlo (MC) method is used to address the uncertainty, is the MC method the MC Dropout? Can the authors state how many times the predictions are called when using this method? Is the model output the average of multiple MC runs? As information on the prediction uncertainty for the ML model is essential for model reliability, I have not found the uncertainty reported by MC in this manuscript, can the authors complement it?

The MC method for uncertainty propagation referred to in our manuscript is not the same as the regularization method „MC dropout". A clarifying explanation was added to the manuscript. The number of MC predictions can be chosen by the user (more predictions → better statistical assessment of the uncertainty). The model output is not the average of multiple MC runs. The MC routine is only used for the assessment of the input uncertainty propagation.

The input uncertainty from the MC routine was not reported for the following reason: In „bulk" comparisons, such as shown in our manuscript, the model output is dominated by the usual machine learning uncertainties (e.g. the model's aleatoric and epistemic error). The input uncertainty can be a relevant error source in the prediction of individual profiles. Then, the relative input uncertainty is approximately equal to the relative uncertainty of the input $NO_2$ VCD, which is typically on the scale of 30 %, but can be estimated more accurately in individual predictions.

The following sentence was added to sect. 4.1: *Because the $NO_2$ VCD is the dominant input variable of NitroNet, and acts essentially as a scaling factor for the predicted $NO_2$ profiles, the relative prediction uncertainty is approximately equal to that of the $NO_2$ VCD (here: 30 % - 60 %).*

8.  Lines 103, 148: How does the "43 terrain-following pressure levels" enable the NitroNet model to output "186 levels"?

The 43 terrain-following pressure levels of WRF-Chem correspond to slightly different altitudes above ground, depending on location and time. Therefore, „altitude" is not a discrete, but continuous variable in the training set. NitroNet can return the $NO_2$ concentration at arbitrary altitudes within the troposphere, as explained in l. 143-149.

The following paragraph was added to sect. 3: *Although WRF-2019 is resolved on 43 vertical pressure levels, these correspond to different altitudes above ground across the spatio-temporal model domain. Therefore, NitroNet can be trained to predict the $NO_2$ concentration at arbitrary tropospheric altitudes.*

9.  Line 187: Can the authors state the total number of parameters or trainable parameters in NitroNet?

A neural network with 8 layers of 326 neurons each has approximately $8 \times 326^2 + 8 \times 326 \approx 850000$ trainable parameters. The first term is due to the network's connection weights, and the second corresponds to the neuron biases. The exact number can vary slightly, because some model components have trainable parameters (e.g. the PReLU activation adds another 8).

The following sentence was added to sect. 3.2: *The neural network has 8 hidden layers with 326 neurons each, corresponding to approximately 850000 trainable parameters.*

10. Section 3.4: Will the out-of-distribution treatment be applied to the model application process or just to the training process? The marginal probability density distributions pxi(x) are calculated on the filtered training set which has only 7% data remaining. If the OOD treatment is also applied to the application process, will many instances be treated as OOD?

Yes, the OOD treatment is applied at prediction time. How many instances are recognized as OOD depends on the OOD threshold (here: 0.15) and the input data, whose values may vary depending on season and location.

The following sentence was added to sect. 3.4: *The described method is applied exclusively at prediction time. The amount of features affected depends mainly on the season and location of the input data.*

11. Lines 238-240: As the NitroNet only has one output neuron for the NO2 concentration, how can an additional output for the ratio F be generated?

The prediction of $F$ is achieved by training the neural network on the $NO_2$ targets, and then subsequently training a copy of it on $F$ targets. The two neural networks are then merged into a single model object. In NitroNet's code interface, the user can pass a parameter `predict_F_values` whose value (True / False) decides whether the model internally uses the $NO_2$ target network, or the $F$ target network.

The following sentence was added to sect. 3.5: *Internally, this additional output is achieved by instantiating a second identical neural network, trained on the F targets from WRF-2019 instead of the NO2 targets.*

12. Lines 318-320: It is suggested to add a validation experiment for 2019 only on the still valid stations in 2022. Considering the significantly reduced number of valid stations in Italy, the difference in the statistics shown in the manuscript makes less sense.

Figure D1 below shows a version of Fig. 5, e.g. a validation against AirBase measurements of the year 2019, where only stations still valid in 2022 are shown.

[Figure]

**Figure D1.** Like Fig. 5 of the manuscript (a comparison of WRF-Chem and NitroNet to AirBase observations of surface $NO_2$ for the year 2019) but with all AirBase instruments removed, that were no longer flagged as „valid" in the year 2022.

Note that this results in slightly fewer stations than shown in Fig. 7b, because not all stations valid in 2022 were available and/or valid in 2019. However, the stations identified as possibly problematic in the discussions of sect. 4.2.1 (e.g. in northern Italy), are removed as expected. The results show slightly better agreement than Fig. 5. However, the differences are small and do not significantly impact the discussions / results of sect. 4.1 and 4.2.1. Although it might be interesting in the context of our manuscript, the method of classifying stations from 2019 based on criteria from 2022 is also questionable. For example, stations are flagged as „invalid", if their measured values are lower than their detection limit. This could well be the case in 2022, but not in 2019, if the local pollution levels shrunk accordingly from 2019 to 2022. Therefore, we

argue that the AirBase station classification of 2019 should not be influenced by posterior knowledge from 2022. For these reasons, we gladly provide the figure in the interactive discussion, but prefer not to add it to the manuscript.

> 13. Figure 8: Please clarify based on which temporal scale (every scatter point or the monthly-mean) these statistics are calculated.

The statistics are calculated based on the monthly-mean scatter points.

The following sentence was added to caption of Figure 8: *(…) and were computed based on the monthly-mean scatter points.*

> 14. Figure 9: "within a radius of 5 km were drawn at 0 m altitude" needs a reference.

This is not a reference to literature. The sentence means: „For each MAX-DOAS station, we have checked for AirBase stations within a radius of 5 km. Then, their measured $NO_2$ concentrations were drawn in the diagrams of Figure 9."

The following sentence was added to sect. 4.2.2: *Additionally, colocated measurements from in situ measurements (within a radius of 5 km) were drawn in the corresponding subplots of Fig. 9.*

> 15. Lines 507-509: The transformer model could be also considered (See https://www.microsoft.com/en-us/research/blog/introducing-aurora-the-first-large-scale-foundation-model-of-the-atmosphere/).

We thank the Referee for this additional suggestion.

The following sentence was added to sect. 5: *This is motivated by the recent advancements in machine learning based weather prediction (e.g. the Aurora model, based on vision transformers and encoder-decoder mechanisms, see Bodnar et al. (2024)).*

Technical corrections:

> 1. Line 18: "geographic domain" should be modified as "geographic and temporal domain". Changed as proposed.

> 2. Please pay attention to writing out acronyms at their first occurrence (e.g., TROPOMI, LIDAR, CAMS, etc). We have perused the manuscript and written out acronyms at their first occurrence after the abstract.

> 3. Line 28 "10 ug/m3": Reference is needed.

The following reference was added:

World Health Organization: WHO global air quality guidelines: particulate matter (PM2.5 and PM10), ozone, nitrogen dioxide, sulfur dioxide and carbon monoxide. 2021.

> 4. Line 44 "to be the main cause": "to be one of the main causes". Changed as proposed.
> 5. Line 69 "CAMS": "CAMS regional". Changed as proposed.

**References:**

Douros, J., Eskes, H., van Geffen, J., Boersma, K. F., Compernolle, S., Pinardi, G., Blechschmidt, A.-M., Peuch, V.-H., Colette, A., and Veefkind, P.: Comparing Sentinel-5P TROPOMI NO2 column observations with the CAMS regional air quality ensemble, Geosci. Model Dev., 16, 509–534, https://doi.org/10.5194/gmd-16-509-2023, 2023.

Kuhn, L., Beirle, S., Kumar, V., Osipov, S., Pozzer, A., Bösch, T., Kumar, R., and Wagner, T.: On the influence of vertical mixing, boundary layer schemes, and temporal emission profiles on tropospheric NO2 in WRF-Chem – comparisons to in situ, satellite, and MAX-DOAS observations, Atmos. Chem. Phys., 24, 185–217, https://doi.org/10.5194/acp-24-185-2024, 2024.

Tirpitz, J.-L., Frieß, U., Hendrick, F., Alberti, C., Allaart, M., Apituley, A., Bais, A., Beirle, S., Berkhout, S., Bognar, K., Bösch, T., Bruchkouski, I., Cede, A., Chan, K. L., den Hoed, M., Donner, S., Drosoglou, T., Fayt, C., Friedrich, M. M., Frumau, A., Gast, L., Gielen, C., Gomez-Martín, L., Hao, N., Hensen, A., Henzing, B., Hermans, C., Jin, J., Kreher, K., Kuhn, J., Lampel, J., Li, A., Liu, C., Liu, H., Ma, J., Merlaud, A., Peters, E., Pinardi, G., Piters, A., Platt, U., Puentedura, O., Richter, A., Schmitt, S., Spinei, E., Stein Zweers, D., Strong, K., Swart, D., Tack, F., Tiefengraber, M., van der Hoff, R., van Roozendael, M., Vlemmix, T., Vonk, J., Wagner, T., Wang, Y., Wang, Z., Wenig, M., Wiegner, M., Wittrock, F., Xie, P., Xing, C., Xu, J., Yela, M., Zhang, C., and Zhao, X.: Intercomparison of MAX-DOAS vertical profile retrieval algorithms: studies on field data from the CINDI-2 campaign, Atmos. Meas. Tech., 14, 1–35, https://doi.org/10.5194/amt-14-1-2021, 2021.

**Authors' reply to RC2 by Anonymous Referee #2**

We would like to thank Anonymous Referee 2 for the constructive and encouraging review of our manuscript. Please find our point-by-point replies below. The Referee's comments are printed in blue, and our replies in black.

Review of Kuhn et al., "NitroNet- A deep-learning NO2 profile retrieval prototype for the TROPOMI satellite instrument"

Reviewer suggestion: minor revisions.

This paper presents a new NO2 retrieval model to produce vertical profiles from satellite observations, using a machine learning approach. In my opinion, this is an impressive piece of work, thoroughly explained, well executed, and producing impressive results. The results of NitroNet comparisons to vertical columns and surface values, within and outside of the training times/regions, shows very good promise.

I think that the main weakness of the paper lies in the challenge of verifying the NO2 vertical profiles, not just columns and surface values. This is inherent to the point of the paper of course, i.e., that NO2 vertical profile measurements are sparse. The authors tackle this by comparison to the FRM4DOAS MAX-DOAS dataset, and results are promising. I think it would improve the paper to include comparison to more MAX-DOAS datasets if possible, outside the European domain and over more seasons. Perhaps this could be achieved by looking at a few discrete layers in the profile, not necessarily full profile comparison plots. I also think the authors should consider whether verification against cloud-sliced NO2 data, or aircraft campaign NOx measurements, are an option to demonstrate the capability of NitroNet to provide information on free- and upper-tropospheric NO2 tropospheric profile.

We acknowledge, that the validation against $NO_2$ profile observations is of high importance. However, the required reference data are hard to obtain. For example:

- $NO_2$ profiles from cloud slicing are usually reported on coarse spatio-temporal grids due to averaging (e.g. seasonal means with 1° x 1° horizontal resolution and 5 tropospheric layers, see Marais et al., 2021).

- A reasonable validation against aircraft measurements would require data recorded on the central European domain after TROPOMI went operational (2017). There are datasets which meet these criteria, (see e.g. Riess et al., 2023; Brenninkmeijer et al., 2007), but these are just as sparse as the FRM4DOAS measurements used in our manuscript.

- $NO_2$ sonde measurements are equally sparse, and should be considered immature compared to other measurement methods.

The possibility of using such reference data for validations in the future was added to the outlook. The following sentence was added to sect. 5: *$NO_2$ profile observations from cloud-slicing (see e.g. Marais (2021)) or aircraft measurements (see e.g. Riess (2023); Brenninkmeijer (2007)) may be used for further validation of NitroNet at various altitudes.*

Based on the Referee's suggestions, the validation against FRM4DOAS profiles was extended. More details are given further below in the reply to the Referees comment referring to l. 425 at the bottom.

I have listed some specific minor revisions below.

Introduction:

It is worth mentioning that there are methods of determining some vertically-resolved NO2 information from satellite observations, e.g. cloud-slicing, and also there are aircraft campaigns providing vertically-resolved NOx information.

See the answer above. Cloud slicing was added to the introduction. Aircraft measurements were already

mentioned there.

The following paragraph was added to sect. 1: *Although further measuring platforms (e.g. sondes, aircraft) and methods (e.g. Light Detection and Ranging instruments (LIDAR), or "cloud-slicing") exist, these are not routinely deployed (see e.g. Sluis et al. (2010); Bourgeois et al. (2022); Lange et al. (2023); Riess et al. (2023); Volten et al. (2009); Berkhout et al. (2018); Su et al. (2021), Marais et al.(2021)). Particularly aircraft measurements and cloud slicing are appreciated for resolving along the vertical axis, although at lower spatio-temporal resolutions (e.g. cloud slicing: seasonal means with 1° × 1° horizontal resolution and 5 tropospheric layers, see Marais et al. (2021)) or sparse spatio-temporal coverage (aircraft measurements).*

You mention that TROPOMI NO2 relies on a priori profiles, but it is also worth noting in your initial comments that the same is true for MAX-DOAS NO2 vertical profiles.

Please note, that only MMF (not MAPA) depends directly on an a priori profile. MAPA depends implicitly on a priori assumptions, e.g. in the form of the profile parametrization.

The following sentence was added to sect. 1: *Additionally, the commonly used retrieval algorithms suffer from significantly reduced sensitivity at higher altitudes (> 2 km), and depend on a priori assumptions.*

Line 89: 'cannot' rather than 'can not', and later in the sentence I think you mean 'inherent to the training data' not ‚immanent…'

We have replaced this occurrence (and several others) of „can not" with „cannot". Also, „immanent" was replaced with „inherent".

Line 110: it would be helpful to the reader to include a brief comment on why the O3 VCDs are included in NitroNet.

The following sentence was added to sect. 2.2: *Additionally, although much less influential, total $O_3$ VCDs are used, assuming they are also informative of the tropospheric $O_3$ column, and thus of the tropospheric $NO_x$ photochemistry.*

Line 137: MAX-DOAS measurements are strongly influenced by clouds. You mention the filtering of clouds by virtue of the selected TROPOMI QA flag: is there a similar filtering for cloudy results for FRM4DOAS MAX-DOAS results?

According to Beirle et al. (2019), MAPA does not provide automatic cloud flagging yet. However, MAPA provides three quality flags („valid", „warning", „error"), which were also shown to be sensitive to cloud effects. In our analysis, we removed all MAPA profiles flagged with „error". No other filter criteria were used.

The following sentence was added to sect. 2.3: *All profiles flagged as "erroneous" by MAPA were discarded. Note, that although MAPA does not support automatic cloud filtering yet, the described „error" flagging was shown to be sensitive to cloud effects, as well (see Beirle et al. (2019)).*

Line 176: A reference for Shapley scores would be good here.

We have added the following reference:

Štrumbelj, E., Kononenko, I. Explaining prediction models and individual predictions with feature contributions. Knowledge and Information Systems 41, 647–665 (2014). https://doi.org/10.1007/s10115-013-0679-x.

Line 191: This statement is a little unclear to me: 'The learning rate was halved whenever training progress had stalled over several epochs'. Perhaps you could clarify?

Close to the loss minima, training can stall if the learning rate is chosen too large. This is because the parameter updates can overshoot, thereby missing the ideal solution to the optimization problem. This can be solved by using a learning rate scheduler, which decreases the learning rate upon stagnation of the loss. In our routine, the learning rate was halved, whenever the validation loss had not decreased over 20

epochs. More information on learning rate schedulers can be found e.g. in the torch documentation (`https://pytorch.org/docs/stable/generated/torch.optim.lr_scheduler.ReduceLROnPlateau.html`).

The following paragraph was added to sect. 3.2: *In order to reduce early stagnation of the training process as a result of too large learning rates, a simple learning rate scheduler was used (*ReduceLROnPlateau*, see Paszke et al. (2019)). The learning rate was halved whenever the training progress, as measured by the validation loss, had stalled over several epochs (meaning full iterations over the training set).*

**Line 211: Is the low bias you mentioned improved or worsened if the filtering criteria are relaxed from the tuned DVCD and DPBLH?**

The bias is given in reference to training the neural network on un-filtered data. Relaxation of the filter criteria would lead to a reduced bias. Note, that the bias can be immediately corrected for, because it is already quantified during training (before „prediction time"), which is already mentioned in the manuscript.

**Line 251: high NO2 in the upper troposphere is also linked to long lifetime of NOx reservoirs, lightning and subsidence from the stratosphere.**

The following sentence was added to sect. 4.1: *(…) which could be linked e.g. to aircraft emissions, decay of $NO_x$ reservoirs, lightning, or stratosphere-troposphere exchange.*

**Line 254: Could you provide a brief comment on why the model performs better at high NO2 concentration than low? Is this largely due to the better agreement in the lower troposphere/more polluted layers?**

The following paragraph was added to sect. 4.1: *The relative prediction errors are smaller at higher $NO_2$ concentrations. This is because the high $NO_2$ concentrations at the surface are more strongly correlated to the $NO_2$ VCD, which is the main model input. Vice versa, the correlation is weaker in higher layers, where the concentration tends to be lower. Therefore, the combined input variables are more descriptive of the lower, more polluted layers, and allow the neural network to make a more precise prediction.*

**Figure 5: I presume that the WRF-Chem comparison to Airbase is achieved with in-situ bias correction (F factor) calculated by WRF-Chem, and that the NitroNet comparison is achieved with F calculated by NitroNet? How well do the F factors agree between WRF-Chem and NitroNet? Could any discrepancies in F factor help explain any of the observed in-situ NO2 biases in Fig 5?**

The comparison is made as described by the Referee. When training NitroNet on the *F* targets, a relative test error (note: not a bias) of 5 % was determined (this was already mentioned in sect. 3.5). Therefore, discrepancies in *F* are most likely not the reason for the observed biases.

The following paragraph was added to sect. 4.1: *As mentioned before, NitroNet is able to reproduce the $NO_z$ correction factors of WRF-Chem with a relative precision of $\pm$ 5 % and no bias. Due to the good agreement between WRF-Chem and NitroNet in this regard, the prediction of the $NO_z$ correction factors cannot explain the low-biases observed in Fig. 5.*

**Figure 9: Is it possible to show the standard deviation of the mean monthly profiles for each technique? It would be interesting to know how significant the profile differences are given the in relation to the variability across the month. Just to clarify, have you only taken MAX-DOAS profiles from FRM4DOAS at the TROPOMI overpass time?**

The standard deviations were added to Fig. 9 as requested by the Referee, and an explanatory sentence was added to its caption. The updated Fig. 9 is also shown below.

The following sentence was added to sect. 4.2.2: *A temporal threshold of 60 minutes is used, meaning that each NitroNet $NO_2$ profile is associated with the average over all colocated MAX-DOAS profiles recorded within 60 minutes of the corresponding satellite overpass.*

[Figure]

**Figure 9.** Comparison of monthly-mean FRM₄DOAS NO₂ profiles against NitroNet profiles (May 2022). The monthly standard deviations of the profiles are drawn as shaded regions in the background. Where available, colocated AirBase measurements of the surface NO₂ concentration within a radius of 5 km were drawn at 0 m altitude.

Line 375: You say in relation to profiles with elevated layers of NO2 that 'NitroNet is unable to reproduce this profile type, most likely because the training dataset contains very few corresponding examples'. Is this something that can be rectified? In principal, or even better if you're able to show it, is it possible to provide more elevated layer examples in the synthetic training data to address this problem?

At the moment, we cannot overcome this obstacle. As described in our preceding publication (Kuhn et al., 2024), the WRF-Chem model struggles to reproduce such elevated layers in many cases (except close to very strong point sources, e.g. power plants). Because NitroNet is trained on WRF-Chem data, it suffers from the same limitations.

In the future, we can attempt to improve NitroNet's profile diversity by

- producing WRF-Chem training data on significantly higher resolution

- attempting to train NitroNet with a weighted training set, where profiles with elevated layers are given larger weights

The following sentence was added to sect. 4.2.2: *As shown in Kuhn et al. (2024), the WRF-Chem model, which provides NitroNet's training data, also struggles to reproduce elevated layers in some locations.*

The following sentence was added to sect. 5: *Similarly, NitroNet could benefit from training data of higher horizontal resolution, which might improve its ability to reproduce more complex $NO_2$ profile shapes, e.g. with elevated layers.*

Line 400-401: There are a number of outstanding research questions related to NOx over the oceans, for example the contribution of ship emissions in the lower troposphere, and the role of lightning in upper tropospheric NOx over the ocean. Is your hypothesis here that NitroNet performs worse over the oceans because the model gets ship NOx emissions wrong, biasing your training set? Rather than state that the oceanic regions are less relevant, it would be good to understand your thoughts on how NitroNet could be improved over the oceans.

We acknowledge the importance of $NO_2$ retrievals over water in relation to the research questions mentioned by the Referee. The relevance of oceanic regions was emphasized in the outlook and the paragraph in question was changed accordingly.

To clarify, our hypothesis is the following: Fig. 10b and Fig. 12a show an abrupt jump in NitroNet's prediction errors at the land-water boundaries. However, this is not observed in Fig. 4. If NitroNet's predictions work well on the training domain, but poorly on foreign domains, it indicates that the foreign domains are characterized by qualitative differences in the combination of features (inputs) and targets (outputs). Such differences could be caused, e.g. by an unrepresentative amount of shipping routes on the training domain.

The following paragraph was added to sect. 4.3: *The most likely explanation is that the training dataset does not contain enough representative examples of $NO_2$ profiles over water. The water regions of the training set must be assumed less representative, e.g. because they are pervaded by unusually many shipping routes, which may lead NitroNet to overestimate $NO_2$ over more remote water bodies.*

The following paragraph was added to sect. 5: *In particular, it might also help to resolve the prediction errors over water, which could be useful in addressing some of the outstanding research questions related to $NO_2$ over the oceans (e.g. the contribution of ship emissions and lightning to the lower / upper troposphere).*

Line 425: In terms of seasonal performance of the vertical profiling capability, it would be really valuable to assess NitroNet against the FRM4DOAS network over seasonal timescales. Seasonal comparison at a few specific altitudes, e.g. 0, 1, 3 km, would give an indicator of whether NitroNet consistently achieves its aim of providing NO2 vertical profiles.

We reply to this comment together with the Referee's suggestions in the initial general comment above.

We acknowledge, that the comparison to $FRM_4DOAS$ data should be more detailed, with specific focus on the assessment of NitroNet's ability to predict realistic $NO_2$ profiles. As suggested by the Referee, an evaluation at individual altitudes over the period of one year was added to the manuscript, and is also shown below in Fig. 14. The altitude ranges (0 - 200 m, 200 - 400 m, 400 - 600 m, 600 - 1000 m, 1000 m - 2000 m) were chosen based on the limited vertical sensitivity of the MAX-DOAS retrievals beyond.

The following paragraphs were added to sect. 4.3:

*Figure 14 shows a full-year evaluation of NitroNet against $NO_2$ concentrations from $FRM_4DOAS$ in selected altitude ranges. For this analysis, NitroNet's average bias (left panel) and absolute error (right panel) over all previously shown $FRM_4DOAS$ instruments were computed for a full year of data, with either MMF or MAPA used as reference. Each subplot of Fig. 14 is restricted to a specific altitude range (0 - 200 m, 200 - 400 m, 400 - 600 m, 600 - 1000 m, 1000 - 2000 m). In the lowest evaluation layer, at 0 - 200 m, there is particularly good agreement between MAPA and MMF, with NitroNet biases between -70 % and +20 % over the course of the year. Here, a similar tendency as in Fig. 13 can be observed, with low biases occurring during winter, and high biases during summer. The summertime high biases are of similar magnitude than in the comparison to TROPOMI VCDs and AirBase surface measurements (approximately + 15 % vs. + 23 %, and*

*+ 10 %, respectively). Particularly in the higher layers, the validation against MMF yields far lower mean biases, mostly in the range from -30 % to + 30 %, while the validation against MAPA result in larger biases of ~ 100 % at 600 - 1000 m, and ~ 200 % at 1000 - 2000 m. This owes to the steeper vertical concentration gradients of the MAPA profiles due to their assumed profile shape, and aligns well with the profiles shown in Fig. 9. The large relative biases of NitroNet in relation to MAPA might appear concerning at first, and should be put into perspective based on the following considerations:*

*First, it is hard to assess, which of the two retrieval algorithms yields more trustworthy results. Although conceptionally different, MAPA and MMF both suffer from increasingly poor sensitivity at higher altitudes. This is also the case here, as exemplified by the MMF averaging kernels shown in Fig. C6, which indicate an effective vertical sensitivity of up to 1.5 km in Heidelberg, May 2022. In consequence, the retrieval results are considerably affected by a priori assumptions. In the case of MMF, an a priori profile is taken from a WRF-Chem simulation over Mexico (see Friedrich et al. (2019)), which might be entirely unrepresentative of the central European domain investigated here. Parametrized retrievals such as MAPA do not require a priori profiles, which is an advantage in this context. Nonetheless, MAPA still depends on other a priori assumptions, e.g. in the form of the assumed profile shape by the choice of parametrization. In particular, the exponential tail of the MAPA profiles towards higher altitudes, which is the dominant characteristic here, is prescribed.*

*Second, computing the relative biases of NitroNet involves division of the absolute errors by the $NO_2$ concentrations of MMF, and MAPA, respectively. In the case of MAPA, these can be considerably small (e.g. ~ 0.1 · $10^{10}$ molec. $cm^{-3}$ for 1000 - 2000 m, see Fig. 9 for reference), for the reasons discussed above. Thereby, even moderate absolute errors (see right-side panel of Fig. 14) can result in large relative biases. Thus, the assessment of model performance by means of the prediction biases is informative in the lowest 3 evaluation layers (up to 600 m), but not beyond.*

*Another important finding of Fig. 14 is that the seasonal trends observed in Fig. 13 are represented in the lowest layer (0 - 200 m), but not the higher ones. This indicates, that the seasonal biases of NitroNet (and the underlying WRF-Chem training data) might be rooted in the lower regions of the troposphere.*

We appreciate the reviewer's suggestion to include FRM4DOAS data from other regions of the world as well. However, the remaining FRM4DOAS instruments are located far away from the central European training domain of NitroNet (e.g. in Ny-Alesund, Norway). As shown in the manuscript, NitroNet's prediction quality can vary under such conditions. Many of these instruments are also operated in remote locations and / or have no colocated AirBase measurements. A validations against these instruments would require considerable additional efforts, introduce new uncertainties, and most likely contribute little to the overall assessment of NitroNet's performance.

[Figure]

**Figure 14.** Seasonal evaluation of NitroNet against NO$_2$ concentrations from the FRM$_4$DOAS dataset. The left panel shows NitroNet's monthly-mean biases averaged over all available MAX-DOAS instruments. The right panel shows the corresponding absolute concentration errors. Each subplot refers to a specific altitude range, namely **(a)** 0 - 200 m, **(b)** 200 - 400 m, **(c)** 400 - 600 m, **(d)** 600 - 1000 m, and **(e)** 1000 - 2000 m.

Figure 13: I may be missing something here, but I'm unsure how the monthly mean correlation coefficients can be almost all above the daily mean correlation coefficients, and the monthly mean RMSE can often be below all the daily RMSE values for a given month (e.g. Apr-Jul 2022)?

In this context, it is important to distinguish between the monthly mean of a diagnostic, and a diagnostic computed on monthly means. Here, „diagnostic" refers to bias, RMSE, or correlation coefficient.

Figure 13 shows the diagnostics computed on monthly-mean data. It does not show the monthly means of the diagnostics computed on daily data. Monthly-mean data has significantly reduced noise compared to unaveraged data, resulting in larger correlation coefficients and lower RMSE. The bias is unaffected by this, because it is insensitive to (centered) noise.

We wish to keep the evaluation this way, and refer to our reply to RC 1. Please note, that the monthly means of the diagnostics computed on daily data can be directly obtained from Fig. 13 as shown. This is not the case for the diagnostics computed on monthly means.

**References:**

Marais, E. A., Roberts, J. F., Ryan, R. G., Eskes, H., Boersma, K. F., Choi, S., Joiner, J., Abuhassan, N., Redondas, A., Grutter, M., Cede, A., Gomez, L., and Navarro-Comas, M.: New observations of NO2 in the upper troposphere from TROPOMI, Atmos. Meas. Tech., 14, 2389–2408, https://doi.org/10.5194/amt-14-2389-2021, 2021.

Riess, T. C. V. W., Boersma, K. F., Van Roy, W., de Laat, J., Dammers, E., and van Vliet, J.: To new heights by flying low: comparison of aircraft vertical NO2 profiles to model simulations and implications for TROPOMI NO2 retrievals, Atmos. Meas. Tech., 16, 5287–5304, https://doi.org/10.5194/amt-16-5287-2023, 2023.

Brenninkmeijer, C. A. M., Crutzen, P., Boumard, F., Dauer, T., Dix, B., Ebinghaus, R., Filippi, D., Fischer, H., Franke, H., Frieβ, U., Heintzenberg, J., Helleis, F., Hermann, M., Kock, H. H., Koeppel, C., Lelieveld, J., Leuenberger, M., Martinsson, B. G., Miemczyk, S., Moret, H. P., Nguyen, H. N., Nyfeler, P., Oram, D., O'Sullivan, D., Penkett, S., Platt, U., Pupek, M., Ramonet, M., Randa, B., Reichelt, M., Rhee, T. S., Rohwer, J., Rosenfeld, K., Scharffe, D., Schlager, H., Schumann, U., Slemr, F., Sprung, D., Stock, P., Thaler, R., Valentino, F., van Velthoven, P., Waibel, A., Wandel, A., Waschitschek, K., Wiedensohler, A., Xueref-Remy, I., Zahn, A., Zech, U., and Ziereis, H.: Civil Aircraft for the regular investigation of the atmosphere based on an instrumented container: The new CARIBIC system, Atmos. Chem. Phys., 7, 4953–4976, https://doi.org/10.5194/acp-7-4953-2007, 2007.

Beirle, S., Dörner, S., Donner, S., Remmers, J., Wang, Y., and Wagner, T.: The Mainz profile algorithm (MAPA), Atmos. Meas. Tech., 12, 1785–1806, https://doi.org/10.5194/amt-12-1785-2019, 2019.

Kuhn, L., Beirle, S., Kumar, V., Osipov, S., Pozzer, A., Bösch, T., Kumar, R., and Wagner, T.: On the influence of vertical mixing, boundary layer schemes, and temporal emission profiles on tropospheric NO2 in WRF-Chem – comparisons to in situ, satellite, and MAX-DOAS observations, Atmos. Chem. Phys., 24, 185–217, https://doi.org/10.5194/acp-24-185-2024, 2024.